# Do We Need All the Synthetic Data? Targeted Image Augmentation via Diffusion Models

**Dang Nguyen**[*†]   **Jiping Li**[*†]   **Jinghao Zheng**[*‡]   **Baharan Mirzasoleiman**[†]
[†]University of California Los Angeles (UCLA)   [‡]Ecole Polytechnique Federale de Lausanne (EPFL)

## Abstract

Synthetically augmenting training datasets with diffusion models has become an effective strategy for improving the generalization of image classifiers. However, existing approaches typically increase dataset size by 10–30× and struggle to ensure generation diversity, leading to substantial computational overhead. In this work, we introduce TADA (**TA**rgeted **D**iffusion **A**ugmentation), a principled framework that selectively augments examples that are not learned early in training using faithful synthetic images that preserve semantic features while varying noise. We show that augmenting only this targeted subset consistently outperforms augmenting the entire dataset. Through theoretical analysis on a two-layer CNN, we prove that TADA improves generalization by promoting homogeneity in feature learning speed without amplifying noise. Extensive experiments demonstrate that by augmenting only 30–40% of the training data, TADA improves generalization by up to 2.8% across diverse architectures including ResNet, ViT, ConvNeXt, and Swin Transformer on CIFAR-10/100, TinyImageNet, and ImageNet, using optimizers such as SGD and SAM. Notably, TADA combined with SGD outperforms the state-of-the-art optimizer SAM on CIFAR-100 and TinyImageNet. Furthermore, TADA shows promising improvements on object detection benchmarks, demonstrating its applicability beyond image classification. Our code is available at `https://github.com/BigML-CS-UCLA/TADA`.

## 1 Introduction

Data augmentation has been essential to obtaining state-of-the-art in image classification tasks. In particular, adding synthetic images generated by diffusion models (Rombach et al., 2022; Nichol et al., 2021; Saharia et al., 2022) improves the accuracy (Azizi et al., 2023) and effective robustness (Bansal & Grover, 2023) of image classification, beyond what is achieved by weak (random crop, flip, color jitter, etc) and strong data augmentation strategies (PixMix, DeepAugment, etc) (Hendrycks et al., 2021; 2022) or data augmentation using traditional generative models (Goodfellow et al., 2020; Brock et al., 2018). Existing works, however, generate synthetic images by conditioning the diffusion model on class labels (Bansal & Grover, 2023; Azizi et al., 2023), or noisy versions of entire training data (Zhou et al., 2023). Unlike weak and strong augmentation techniques, data augmentation techniques based on diffusion models often struggle to ensure diversity and increase the size of the training data by up to 10× (Azizi et al., 2023) to 30× (Fu et al., 2024) to yield satisfactory performance improvement. This raises a key question:

*Does synthetically augmenting the full data yield optimal performance? Can we identify a part of the data that outperforms full data, when synthetically augmented?*

At first, this seems implausible as adding synthetic images corresponding to only a part of the training data introduces a shift between training and test data distributions and harms the in-distribution performance. However, recent results in the optimization literature have revealed that learning features at a more uniform speed during training improves the generalization performance (Nguyen et al., 2024). This is shown by comparing learning dynamics of Sharpness-Aware-Minimization (SAM) with Gradient Descent (GD) optimizers. SAM is a state-of-the-art optimizer that finds flatter local minima by simultaneously minimizing the value and sharpness of the loss (Foret et al., 2020).

---

[*]Equal contribution

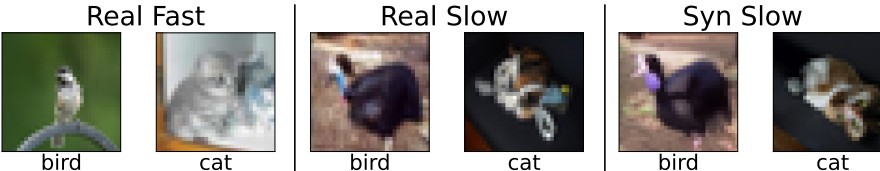

Figure 1: Examples of slow- and fast-learnable images and our *faithful* synthetic images corresponding to slow-learnable examples generated for CIFAR-10. Our synthetic data preserves features in slow-learnable images but replace noise. This amplifies slow-learnable features without magnifying noise. This is difficult to achieve with standard augmentations like random cropping or flipping, highlighting the value of generative augmentation. Additional images are given in Figure 7.

In doing so, SAM learns slow-learnable features faster than GD and achieves superior performance. This suggests that augmenting the slow-learnable part of the data to accelerate their learning can improve the generalization performance, despite slight distribution shift. Yet, how to generate such synthetic data remains an open question.

In our work, we provide a rigorous answer to the above question. We propose TADA (**TA**rgeted **D**iffusion **A**ugmentation), a principled framework for selectively augmenting slow-learnable examples using diffusion models. First, by analyzing a two-layer convolutional neural network (CNN), we show that SAM suppresses learning noise from the data, while speeding up learning slow-learnable features. Then, we prove that generating *faithful* synthetic images containing slow-learnable features with different noise effectively speeds up learning such features without causing noise overfitting. To find examples with slow-learnable features, we partition the data to two parts by clustering model outputs early in training and identify the cluster with higher average loss. Then, we generate faithful images corresponding to the slow-learnable examples, by using real data to guide the diffusion process. That is, we add noise to examples that are not learned early in training and denoise them to generate faithful synthetic data (see examples in Figure 1). This enables synthetically augmenting only the slow-learnable part of the data by up to 5x to get further performance improvement. In contrast, upsampling slow-learnable examples—which appears to be a simpler and more intuitive approach—more than once could amplify the noise and significantly harm the performance, suggesting the necessity of using synthetic data. Finally, we prove the convergence properties of training on our synthetically augmented data with stochastic gradient methods.

We conduct extensive experiments training ResNet, ViT, DenseNet, ConvNeXt, and Swin Transformer on CIFAR10, CIFAR100 (Krizhevsky et al., 2009), TinyImageNet (Le & Yang, 2015), and ImageNet (Deng et al., 2009). We show that TADA consistently outperforms both upsampling and full synthetic augmentation, improving SGD and SAM by up to 2.8% while augmenting only 30–40% of the data. Notably, TADA combined with SGD outperforms SAM on CIFAR-100 and TinyImageNet and achieves state-of-the-art performance. Moreover, we validate its scalability on ImageNet with ResNet18 and ResNet50, and demonstrate its effectiveness beyond classification by improving object detection performance. TADA remains effective across different diffusion models and can be seamlessly combined with existing weak and strong augmentation strategies to further boost performance.

## 2 RELATED WORKS

**Generative Models for Augmentation.** There has been a recent surge of studies on synthetic data augmentation using diffusion models. For example, Azizi et al. (2023) applied diffusion models to ImageNet classification, while further studies (Trabucco et al., 2023; He et al., 2023) explored their application in zero- or few-shot settings. Despite promising results, this line of research faces fundamental challenges in achieving diversity, faithfulness, and efficiency. Recent work attempts to overcome this through intricate prompt-conditioning mechanisms, customized embedding optimizations, or multi-stage diffusion processes. For example, DiffuseMix (Islam et al., 2024a) and GenMix (Islam et al., 2024b) use prompt-guided editing with complex mixing strategies to avoid unrealistic artifacts, while Diff-Mix (Wang et al., 2024) balances foreground fidelity and background diversity through inter-class mixup. Diff-II (Wang & Chen, 2025) introduces a novel inversion-circle interpolation and a two-stage denoising process to jointly promote diversity and faithfulness. For fine-grained

image classification, SaSPA (Michaeli & Fried, 2024) preserves structural integrity by conditioning on edges and subject representations, and DiffCoRe-Mix (Islam & AKHTAR, 2025) uses constrained diffusion with negative prompting and hard-cosine filtering to maintain semantic consistency.

Although these approaches improve synthetic image quality and diversity, they typically require generating extremely large synthetic datasets—often $10\times$ to $30\times$ the size of the original data—to achieve meaningful performance gains, making them computationally expensive. A few recent works, such as Boomerang (Luzi et al., 2022) and DiffCoRe-Mix, address this cost by performing local manifold sampling, enabling strong performance with a $1\times$ augmentation ratio. However, these methods still involve substantial system-level complexity and high generation costs.

Our approach departs from this trend by focusing on *which* examples to augment rather than designing increasingly complex generation pipelines. We theoretically and empirically show that augmenting only the 30%–40% of examples that are not learned early in training is sufficient—and often superior—to full-data augmentation. TADA is simple, computationally lightweight, and generator-agnostic. It can be seamlessly combined with state-of-the-art diffusion-based augmentation methods, as demonstrated with both DiffuseMix and Boomerang in our experiments.

**Sharpness-aware-minimization (SAM).** SAM is an optimization technique that obtains state-of-the-art performance on a variety of tasks, by simultaneously minimizing the loss and its sharpness (Foret et al., 2020; Zheng et al., 2021). In doing so, it improves the generalization in expense of doubling the training time. SAM has also been shown to be beneficial in settings such as label noise (Foret et al., 2020; Zheng et al., 2021), out-of-distribution (Springer et al., 2024), and domain generalization (Cha et al., 2021; Wang et al., 2023).

The superior generalization performance of SAM has been contributed to smaller Hessian spectra (Foret et al., 2020; Kaur et al., 2023; Wen et al., 2022; Bartlett et al., 2023), sparser solution (Andriushchenko & Flammarion, 2022), and benign overfitting in presence of weaker signal (Chen et al., 2022). Most recently, SAM is shown to learn features at a more uniform speed (Nguyen et al., 2024). In our work, we show that targeted synthetic data augmentation can improve generalization by making training dynamics more similar to SAM.

## 3 PRELIMINARY

In this section, we introduce our theoretical framework for analyzing synthetic data augmentation with diffusion models. We also discuss SAM's ability in learning features at a more uniform speed.

**Data Distribution.** We adopt a similar data distribution used in recent works on feature learning (Allen-Zhu & Li, 2020; Chen et al., 2022; Jelassi & Li, 2022; Cao et al., 2022; Kou et al., 2023; Deng et al., 2023; Chen et al., 2023) to model data containing two features $\boldsymbol{v}_d, \boldsymbol{v}_e$ and noise patches.

**Definition 3.1** (Data distribution). A data point has the form $(\boldsymbol{x}, y) \sim \mathcal{D}(\beta_e, \beta_d, \alpha) \in (\mathbb{R}^d)^P \times \{\pm 1\}$, where $y \sim \text{Radamacher}(0.5), 0 \leq \beta_d < \beta_e \in \mathbb{R}$, and $\boldsymbol{x} = (\boldsymbol{x}^{(1)}, \boldsymbol{x}^{(2)}, \cdots, \boldsymbol{x}^{(P)})$ contains $P$ patches.

- Exactly one patch is given by the *fast-learnable* feature $\beta_e \cdot y \cdot \boldsymbol{v}_e$ for some unit vector $\boldsymbol{v}_e$ with probability $\alpha > 0$. Otherwise, the patch is given by the *slow-learnable* feature $\beta_d \cdot y \cdot \boldsymbol{v}_d$ for some unit vector $\boldsymbol{v}_e \cdot \boldsymbol{v}_d = 0$.

- The other $P - 1$ patches are i.i.d. Gaussian noise $\boldsymbol{\xi}$ from $\mathcal{N}(0, (\sigma_p^2/d)\boldsymbol{I}_d)$ for some constant $\sigma_p$.

Probability $\alpha$ controls the frequency of feature $\boldsymbol{v}_e$ in the data distribution. The distribution parameters $\beta_e, \beta_d$ characterize the feature strength in the data. $\beta_e > \beta_d$ ensures that the fast-learnable feature is represented better in the population and thus learned faster. The faster speed of learning captures various notions of simplicity, such as simpler shape, larger magnitude, and less variation. Note that image data in practice are high-dimensional and the noises become dispersed. For simplicity, we assume $P = 2$, that the noise patch is orthogonal from the two features, and that summations involving noise cross-terms $\langle \boldsymbol{\xi}_i, \boldsymbol{\xi}_j \rangle$ become negligible.

**Two-layer CNN Model.** We use a dataset $D = \{(\boldsymbol{x}_i, y_i)\}_{i=1}^N$ from distribution 3.1 to train a two-layer nonlinear CNN with activation functions $\sigma(z) = z^3$.

**Definition 3.2** (Two-layer CNN). For one data point $(\boldsymbol{x}, y)$, the two-layer Convolutional Neural Network (CNN) with weights $\boldsymbol{W} = [\boldsymbol{w}_1, \boldsymbol{w}_2, \cdots, \boldsymbol{w}_J] \in \mathbb{R}^{d \times J}$, where $\boldsymbol{w}_j$ is weight of the $j$-th

neuron (filter), has the form:

$$f(\boldsymbol{x}; \boldsymbol{W}) = \sum_{j=1}^{J} \sum_{p=1}^{P} \langle \boldsymbol{w}_j, \boldsymbol{x}^{(p)} \rangle^3 = \sum_{j=1}^{J} \left( \langle \boldsymbol{w}_j, \boldsymbol{\xi} \rangle^3 + y \begin{cases} \beta_d^3 \langle \boldsymbol{w}_j, \boldsymbol{v}_d \rangle^3 & \text{if } \boldsymbol{v}_d \\ \beta_e^3 \langle \boldsymbol{w}_j, \boldsymbol{v}_e \rangle^3 & \text{if } \boldsymbol{v}_e \end{cases} \right) \quad \text{for our } P = 2.$$

**Empirical Risk Minimization.** We consider minimizing the following empirical logistic loss:

$$\mathcal{L}(\boldsymbol{W}) = \frac{1}{N} \sum_{i=1}^{N} l(y_i f(\boldsymbol{x}_i; \boldsymbol{W})) := \frac{1}{N} \sum_{i=1}^{N} \log(1 + \exp(-y_i f(\boldsymbol{x}_i; \boldsymbol{W}))). \tag{1}$$

via (1) sharpness-aware minimization (SAM) (Foret et al., 2020) and (2) gradient descent (GD), whose filter-wise update rules, with some learning rate $\eta > 0$, are respectively given by:

$$\textbf{SAM} : \boldsymbol{w}_j^{(t+1)} = \boldsymbol{w}_j^{(t)} - \eta \nabla_{\boldsymbol{w}_j^{(t)}} \mathcal{L}(\boldsymbol{W}^{(t)} + \rho^{(t)} \nabla \mathcal{L}(\boldsymbol{W}^{(t)})), \text{ where } \rho^{(t)} = \rho / \|\nabla \mathcal{L}(\boldsymbol{W}^{(t)})\|_F, \rho > 0 ,$$

$$\textbf{GD} : \boldsymbol{w}_j^{(t+1)} = \boldsymbol{w}_j^{(t)} - \eta \frac{1}{N} \sum_{i=1}^{N} \nabla_{i, \boldsymbol{w}_j^{(t)}} \mathcal{L}(\boldsymbol{W}^{(t)}) = \boldsymbol{w}_j^{(t)} - \eta \nabla_{\boldsymbol{w}_j^{(t)}} \mathcal{L}(\boldsymbol{W}^{(t)}).$$

Here $\nabla_{\boldsymbol{w}_j^{(t)}} \mathcal{L}(\boldsymbol{W}^{(t)})$ denotes the full gradient w.r.t. filter $\boldsymbol{w}_j$ at iteration $t$, $\nabla_{i, \boldsymbol{w}_j^{(t)}} \mathcal{L}(\boldsymbol{W}^{(t)})$ denotes the per-example gradient for $i \in [N]$, and $\nabla \mathcal{L}(\boldsymbol{W}^{(t)})$ denotes the full gradient matrix.

**High-level idea of SAM.** By perturbing the weights with gradient ascent ($\boldsymbol{\epsilon}^{(t)} = \rho^{(t)} \nabla \mathcal{L}(\boldsymbol{W}^{(t)})$), SAM looks ahead in the *worst* weight direction and forces the training algorithm to escape an unstable (sharp) local minimum. In practice, this leads to more generalizable solutions.

**SAM Learns Features More Homogeneously.** With the above setting, the alignment of $\boldsymbol{v}_d, \boldsymbol{v}_e$ with weights, i.e., $\langle \boldsymbol{w}^{(t)}, \boldsymbol{v}_d \rangle$ and $\langle \boldsymbol{w}^{(t)}, \boldsymbol{v}_d \rangle$, indicate how much they are learned by the CNN at iteration $t$. SAM's (normalized) gradient for the *slow-learnable* feature is larger than GD by a factor of (Nguyen et al., 2024):

$$k = \left( \frac{1 - \rho^{(t)} \beta_d^3 \langle \boldsymbol{w}, \boldsymbol{v}_d \rangle}{1 - \rho^{(t)} \beta_e^3 \langle \boldsymbol{w}, \boldsymbol{v}_e \rangle} \right)^{2/3}. \tag{2}$$

That is, SAM amplifies the slow-learnable feature and learns it faster than GD. In doing so, it learns features at a more homogeneous speed. While Eq. 2 suggests that the empirical choice of $k$ should depend on the relative strength and difficulty of the features, simply upsampling examples with slow-learnable features more than once results in performance degradation, as we confirm in our experiments.

## 4 Learning Features Homogeneously without Overfitting Noise

In this section, we first prove that SAM suppresses learning noise from the data, while promoting homogeneous feature learning. Then, we introduce TADA (**TA**rgeted **D**iffusion **A**ugmentation) and discuss generating synthetic data to amplify features in images without magnifying their noise. This allows amplifying slow-learnable features by $k > 2$ to further boost performance. Finally, we show convergence of training on our synthetically augmented data.

### 4.1 SAM Suppresses Learning Noise from the Data

First, we theoretically analyze how SAM suppresses learning noise in the above setting. Intuitively, as SAM pushes the learning dynamics away from sharp landscapes, it simultaneously helps the model avoid areas where certain noises concentrate. This becomes a natural defense against noise overfitting in high-curvature areas. On the other hand, gradient descent is unaware of local smoothness, so solutions that may sit in a flat, noise-resilient basin.

Fomally, we prove that starting with the same weights $\boldsymbol{W}^{(t)}$, a SAM step suppresses the model's alignment with noise directions more effectively than an equivalent gradient descent step. Let $\boldsymbol{\Phi}$ denote the sets of noises for dataset $D$. Let $\boldsymbol{w}_{j,\boldsymbol{\epsilon}}$ denote the perturbed weights of filter $j$ for SAM. We then define $\mathcal{I}_{j,\boldsymbol{\epsilon},+}^{(t)} = \{ \boldsymbol{\phi}_i \in \boldsymbol{\Phi} : i \in [N], \text{sgn}(\langle \boldsymbol{w}_{j,\boldsymbol{\epsilon}}^{(t)}, \boldsymbol{\phi}_i \rangle) = \text{sgn}(y_i) \}$ and $\mathcal{I}_{j,\boldsymbol{\epsilon},-}^{(t)} = \{ \boldsymbol{\phi}_i \in \boldsymbol{\Phi} :$

$i \in [N]$, $\text{sgn}(\langle \boldsymbol{w}_{j,\boldsymbol{\epsilon}}^{(t)}, \boldsymbol{\phi}_i \rangle) \neq \text{sgn}(y_i)\}$ be the sets of noises where the sign of alignment matches or mismatches the sign of the label. We define $\mathcal{I}_{j,+}^{(t)}$, $\mathcal{I}_{j,-}^{(t)}$ accordingly for each GD weight $\boldsymbol{w}_j^{(t)}$. We measure filter-wise noise learning for a set of noises using the metric **NoiseAlign**$(\mathcal{I}, \boldsymbol{w}_j^{(t)}) = \frac{1}{|\mathcal{I}|} \sum_{\phi \text{ in } \mathcal{I}} |\langle \nabla_{\boldsymbol{w}_j^{(t)}} \mathcal{L}(\boldsymbol{W}^{(t)}), \phi \rangle|$, which intuitively measures how much noise is learned by the model.

The following theorem quantifies how SAM learns noise to a smaller extent compared to GD. For simplicity, we analyze the early training phase. However, our results should hold throughout the training.

**Theorem 4.1.** *With controlled logit terms $l_i^{(t)} = sigmoid(-y_i f(\boldsymbol{x}_i; \boldsymbol{W}^{(t)}))$, large data size $N$, small learning rate $\eta$, and small SAM perturbation parameter $\rho$ (see Appendix A), SAM and GD updates from the same parameters have the following property, early in training:*

1. ***Inert Noises:*** *Alignment with noises that belong to $\mathcal{I}_{j,-}^{(t)}$ and $\mathcal{I}_{j,\boldsymbol{\epsilon},-}^{(t)}$ will get closer to $0$ after each update, so they will not be learned eventually by GD or SAM.*

2. ***Noise Learning:*** *The other noises will continue being learned in the sense that $|\langle \boldsymbol{w}_j, \boldsymbol{\xi}_i \rangle|$ is monotonically increasing. For these noises, SAM slows down noise learning by looking ahead to noise-sensitive (sharp) directions, while GD updates "blindly". The following SAM and GD learning dynamics hold for $\boldsymbol{\xi}_i \in \mathcal{I}_{j,\boldsymbol{\epsilon},+}^{(t)}$ and $\boldsymbol{\xi}_i \in \mathcal{I}_{j,+}^{(t)}$ in terms of noise gradient:*

$$\textbf{SAM: } |\langle \nabla_{\boldsymbol{w}_{j,\boldsymbol{\epsilon}}} \mathcal{L}(\boldsymbol{W}^{(t)} + \boldsymbol{\epsilon}^{(t)}), \boldsymbol{\xi}_i \rangle| = \frac{3}{N} l_i^{(t)} \langle \boldsymbol{w}_j^{(t)}, \boldsymbol{\xi}_i \rangle^2 \left(1 - \frac{3\rho^{(t)}}{N} l_i^{(t)} |\langle \boldsymbol{w}_j^{(t)}, \boldsymbol{\xi}_i \rangle| \|\boldsymbol{\xi}_i\|^2 \right)^2 \|\boldsymbol{\xi}_i\|^2,$$

$$\textbf{GD: } \quad |\langle \nabla_{\boldsymbol{w}_j} \mathcal{L}(\boldsymbol{W}^{(t)}), \boldsymbol{\xi}_i \rangle| = \frac{3}{N} l_i^{(t)} \langle \boldsymbol{w}_j^{(t)}, \boldsymbol{\xi}_i \rangle^2 \|\boldsymbol{\xi}_i\|^2.$$

*Furthermore, on average, the perturbed SAM gradient aligns strictly less with these noises,*

$$\textbf{NoiseAlign}(\mathcal{I}_{j,\boldsymbol{\epsilon},+}^{(t)}, \boldsymbol{w}_{j,\boldsymbol{\epsilon}}^{(t)}) < \textbf{NoiseAlign}(\mathcal{I}_{j,+}^{(t)}, \boldsymbol{w}_{j,\boldsymbol{\epsilon}}^{(t)}).$$

*A special case of this theorem is that with the same initializations $\boldsymbol{W}^{(0)} \sim \mathcal{N}(0, \sigma_0^2)$, nearly half of the noises will not be learned, and SAM in early training prevents overfitting, while GD does not.*

All the proof can be found in Appendix A. Our results are aligned with (Chen et al., 2023) which showed, in a different setting, that SAM can achieve benign overfitting when SGD cannot.

**Remark.** Theorem 4.1 implies that to resemble feature learning with SAM and ensure superior convergence, it is also crucial to avoid magnifying noise when amplifying the slow-learnable feature.

## 4.2 SYNTHETIC DATA AUGMENTATION TO AMPLIFY FEATURES BUT NOT NOISE

Next, we discuss how TADA identifies examples containing slow-learnable features and generates faithful synthetic data to amplify these features without magnifying the noise in the data.

**Identifying slow-learnable examples.** To find slow-learnable features in the data, we find examples that are not learned robustly at the early phase of training. If an example contains at least one fast-learnable feature that is learned by the model early in training, the model relies on such features to lower the loss and potentially correctly classify the example early in training. Thus, examples that only include slow-learnable features can be identified based on loss or misclassification, or by partitioning model outputs to two clusters after a few training epochs and selecting the cluster with the higher average loss. In our experiments, we use clustering to identify examples with slow-learnable features as it yields better performance, as we confirm in our ablation studies. We note that finding examples with slow-learnable features is not the main focus of our work. Our main contribution is characterizing *how* to amplify slow-learnable features without amplifying noise in the data.

**Amplifying slow-learnable features.** Next, we show that generating synthetic data containing slow-learnable features with different noise considerably boosts the generalization performance, while upsampling slow-learnable examples amplifies the noise in the data and harms generalization.

Recall that $D$ is the original dataset with $|D| = N$. We assume exactly $(1 - \alpha)N \in \mathbb{Z}$ samples have only $\boldsymbol{v}_d$, and let $D_U$, $D_G$ be the modified datasets via upsampling and generation with factor $k$ and new size $N_{new} = \alpha N + k(1 - \alpha)N$. For $D_U$, the replicated noises $\{\boldsymbol{\xi}_i : i = \alpha N + 1, \dots, N\}$

introduce a dependence. Additionally, for $D_G$, the generative model will have its own noises $\boldsymbol{\gamma}_i$ (potentially higher) for synthetic data, and we assume the noises are i.i.d., orthogonal to features, and independent from feature noise from some distribution $\mathcal{D}_{\boldsymbol{\gamma}}$.

With similar notations, let $\boldsymbol{\Phi}_G$, $\boldsymbol{\Phi}_U$ denote the multi-sets [1] of all the noises for $D_G$, $D_U$ respectively. We define $\mathcal{I}_{j,+}^{G,(t)} = \{\boldsymbol{\phi}_i \in \boldsymbol{\Phi}_G : i \in [N_{new}], \ \mathrm{sgn}\langle \boldsymbol{w}_j^{(t)}, \boldsymbol{\phi}_i \rangle = \mathrm{sgn}(y_i)\}$, $\mathcal{I}_{j,-}^{G,(t)}$, $\mathcal{I}_{j,+}^{U,(t)}$, $\mathcal{I}_{j,-}^{U,(t)}$ in a similar fashion for $D_G$ and $D_U$ as before.

Next, we show that early in training, upsampling and generation contribute similarly to feature learning, but upsampling accelerates learning noises in $\mathcal{I}_{j,+}^{U,(t)}$, while synthetic generation does not.

**Theorem 4.2** (Comparison of feature & noise learning). *Let $\nabla_{\boldsymbol{w}_j^{(t)}}^U(\mathcal{L}(\boldsymbol{W}^{(t)}))$ and $\nabla_{\boldsymbol{w}_j^{(t)}}^G(\mathcal{L}(\boldsymbol{W}^{(t)}))$ denote the full gradients w.r.t the $j$-th neuron for upsampling and generation at iteration $t$ respectively. When dropping the index $i$, the term is treated as a random variable. Then for one gradient update,*

1. ***Feature Learning & Inert Noises:*** *Both gradients attempt to contribute equally to feature learning and will not eventually learn certain noises (that are those in $\mathcal{I}_{j,-}^{U,(t)}$, $\mathcal{I}_{j,-}^{G,(t)}$).*

2. ***Noise Learning:*** *Upsampling, compared with generation, amplifies learning of noise on the repeated subset by a factor of $k$. In particular, for generation, $\boldsymbol{\phi}_i \in \mathcal{I}_{j,\epsilon,+}^{G,(t)}$, where $\boldsymbol{\phi}_i$ could be $\boldsymbol{\xi}_i$ or the generation noise $\boldsymbol{\gamma}_i$,*

$$|\langle \nabla_{\boldsymbol{w}_j}^G \mathcal{L}(\boldsymbol{W}^{(t)}), \boldsymbol{\phi}_i \rangle| = \frac{3}{N_{new}} l_i^{(t)} \langle \boldsymbol{w}_j^{(t)}, \boldsymbol{\phi}_i \rangle^2 \|\boldsymbol{\phi}_i\|^2.$$

*However, for upsampling, $\boldsymbol{\xi}_i \in \mathcal{I}_{j,+}^{U,(t)}$,*

$$|\langle \nabla_{\boldsymbol{w}_j}^U \mathcal{L}(\boldsymbol{W}^{(t)}), \boldsymbol{\xi}_i \rangle| = \begin{cases} \frac{3}{N_{new}} l_i(t) \langle \boldsymbol{w}_j^{(t)}, \boldsymbol{\xi}_i \rangle^2 \|\boldsymbol{\xi}_i\|^2, & i = 1, \dots, \alpha N \\ \frac{3k}{N_{new}} l_i(t) \langle \boldsymbol{w}_j^{(t)}, \boldsymbol{\xi}_i \rangle^2 \|\boldsymbol{\xi}_i\|^2, & i = \alpha N + 1, \dots, N \end{cases}.$$

*A special case of this theorem is that with the same initializations $\boldsymbol{W}^{(0)} \sim \mathcal{N}(0, \sigma_0^2)$, for all the early iterations $0 \le t \le T$, if the synthetic noises are sufficiently small in the sense that*

$$\mathbb{E}\left[ l_{\boldsymbol{\gamma}}(t) \langle \boldsymbol{w}_j^{(t)}, \boldsymbol{\gamma} \rangle^2 \|\boldsymbol{\gamma}\|^2 \right] < (k+1) \mathbb{E}\left[ l_{\boldsymbol{\xi}}(t) \langle \boldsymbol{w}_j^{(t)}, \boldsymbol{\xi} \rangle^2 \|\boldsymbol{\xi}\|^2 \right],$$

*where $l_{\boldsymbol{\gamma}}(t)$, $l_{\boldsymbol{\xi}}(t)$ denote the logits as random variables under the corresponding noise at iteration $t$,*

*then noises are overfitted less:* $\quad \mathbb{E}\left[ \textbf{NoiseAlign}(\mathcal{I}_{j,+}^{G,(t)}, \boldsymbol{w}_j^{(t)}) \right] < \mathbb{E}\left[ \textbf{NoiseAlign}(\mathcal{I}_{j,+}^{U,(t)}, \boldsymbol{w}_j^{(t)}) \right].$

Theorem 4.2 suggests that TADA can prevent noise overfitting in expectation by synthetically generating faithful images that preserve features in real data with independent noise. In particular, this metric of generation noise does not need to be strictly less than that of data noise during early training; instead, there exists a tolerance factor up to $k + 1$.

**Remark.** For a diffusion model, as long as the noises in the synthetic data $\|\boldsymbol{\gamma}_i\|^2$ are not too large, noise learning under generation is more diffused due to its independence and operates more similarly to SAM, whereas upsampling amplifies learning on the duplicated noise, potentially elevating it to a new feature and increasing the **NoiseAlign** metric.

### 4.3 SUPERIOR CONVERGENCE OF SYNTHETIC DATA AUGMENTATION OVER UPSAMPLING

Next, we show the superior convergence of mini-batch Stochastic Gradient Descent (SGD)—which is used in practice—when training with TADA-generated synthetic augmentation compared to direct upsampling. Consider training the above CNN using SGD with batch size B: $\boldsymbol{w}_j^{(t+1)} = \boldsymbol{w}_j^{(t)} - \eta \frac{1}{B} \sum_{i=1}^{B} \nabla_{i, \boldsymbol{w}_j^{(t)}} \mathcal{L}(\boldsymbol{W}^{(t)})$. The convergence rate of mini-batch SGD is inversely proportional

---

[1] Due to identical noises in $D_U$. The following $\mathcal{I}_{j,+}^{U,(t)}$ and $\mathcal{I}_{j,-}^{U,(t)}$ are also defined via multi-sets to keep their sizes consistent with the generation counterparts, meaning that we count all the replicated noises.

to the batch size (Ghadimi & Lan, 2013). The following theorem shows that upsampling inflates the variance of mini-batch gradients and thus slows down convergence.

Note that it does not directly compare the relative magnitudes of the variances, since $\sigma_G(k)$, $\sigma_U(k)$ depend on datasets. Instead, it quantifies the sources of extra variance:

**Theorem 4.3** (Variance of mini-batch gradients). *Suppose we train the model using mini-batch SGD with proper batch size $B$. Let $\mathbb{E}_{i \in D_G}[\|\boldsymbol{g}_i - \bar{\boldsymbol{g}}_G\|^2] \leq \sigma_G^2(k)$, $\mathbb{E}_{i \in D_U}[\|\boldsymbol{g}_i - \bar{\boldsymbol{g}}_U\|^2] \leq \sigma_U^2(k)$ be the variances of the per-example gradients for generation and upsampling (where $\boldsymbol{g}_i$ is the gradient of the $i$-th data and $\bar{\boldsymbol{g}}$ is the full gradient). Let $\hat{\boldsymbol{g}}_U$ and $\hat{\boldsymbol{g}}_G$ be the mini-batch gradients. We have:*

$$\mathbb{E}_{D_G}[\|\hat{\boldsymbol{g}}_G - \bar{\boldsymbol{g}}_G\|^2] \leq \frac{\sigma_G^2(k)}{B},$$

$$\mathbb{E}_{D_U}[\|\hat{\boldsymbol{g}}_U - \bar{\boldsymbol{g}}_U\|^2] \leq I, \quad where \; I \geq \frac{\sigma_U^2(k)}{B}\left(1 + \frac{k(k-1)(1-\alpha)}{(\alpha + k(1-\alpha))^2}\frac{B}{N}\right).$$

From Theorem 4.3, we see that all the variance for generation solely comes from the per-example variance, potentially getting larger when synthetic images diversify the dataset. However, upsampling induces one extra term that results from repeated noises and unnecessarily inflates the variance due to dependence within the dataset. This is empirically justified in our ablation studies in Section 5.2.

**Corollary 4.4.** *As long as the generation noise is small enough, i.e., $\sigma_G^2(k) \leq \sigma_U^2(k)$, convergence of mini-batch SGD on synthetically augmented data is faster than upsampled data.*

### 4.4 GENERATING FAITHFUL SYNTHETIC IMAGES VIA DIFFUSION MODELS

Finally, we describe how TADA generates *faithful* images for the slow-learnable part of the data. From Theorem 4.2 we know that while the noise in the synthetic data can be larger than that of the original data, it should be small enough to yield a similar feature learning behavior to SAM.

Synthetic image generation with diffusion models (Sohl-Dickstein et al., 2015; Ho et al., 2020; Nichol & Dhariwal, 2021; Yang et al., 2023) involves a forward process to iteratively add noise to the images, followed by a reverse process to learn to denoise the images. Specifically, the forward process progressively adds noise to the data $x_0$ over $T$ steps, with each step modeled as a Gaussian transition: $q(\mathbf{x}_t|\mathbf{x}_{t-1}) = \mathcal{N}(\mathbf{x}_t; \sqrt{1-\beta_t}\mathbf{x}_{t-1}, \beta_t\mathbf{I})$, where $\beta_t$ controls the noise added at each step. The reverse process inverts the forward process, learns to denoise the data, with the goal of recovering the original data $x_0$ from a noisy sample $x_T$: $p_\theta(\mathbf{x}_{t-1}|\mathbf{x}_t) = \mathcal{N}(\mathbf{x}_{t-1}; \mu_\theta(\mathbf{x}_t, t), \Sigma_\theta(\mathbf{x}_t, t))$, where mean $\mu_\theta$ is conditioned on the sample at the previous time step and variance $\Sigma_\theta$ follows a fixed schedule.

To ensure generating images that are faithful to real data, we use the real images as guidance to generate synthetic images. Specifically, while using the class name (e.g., "a photo of a dog") as the text prompt, we also incorporated the original real samples as guidance. More formally, instead of sampling a a pure noisy image $x_T \sim \mathcal{N}(0, I)$ as the initialization of the reverse path, we add noise to a reference (real) image $x_0^{ref}$ such that the noise level corresponds to a certain time-step $t_*$. Then we begin denoising from time-step $t_*$, using an open-source text-to-image model, e.g. GLIDE (Nichol et al., 2021), to iteratively predict a less noisy image $x_{t-1}(t = T, T-1, \cdots, 1)$ using the given text prompt $l$ and the noisy latent image $x_t$ as inputs. This technique enables produce synthetic images that are similar, yet distinct, from the original examples, and has been successfully used for synthetic image generation for few-shot learning (He et al., 2023). Pseudocode of TADA is illustrated in Appendix E Alg. 1.

**Generality.** While we describe TADA using diffusion models, our targeted augmentation strategy is compatible with other diffusion models and synthetic generation pipelines; as shown in our experiments, it can be instantiated with different generative backbones.

## 5 EXPERIMENT

In this section, we evaluate the effectiveness of our synthetic augmentation strategy on various datasets and model architectures. We also conduct an ablation study on different parts of our method.

**Base training datasets.** We use common benchmark datasets for image classification including CIFAR10, CIFAR100 (Krizhevsky et al., 2009), Tiny-ImageNet (Le & Yang, 2015), ImageNet (Deng

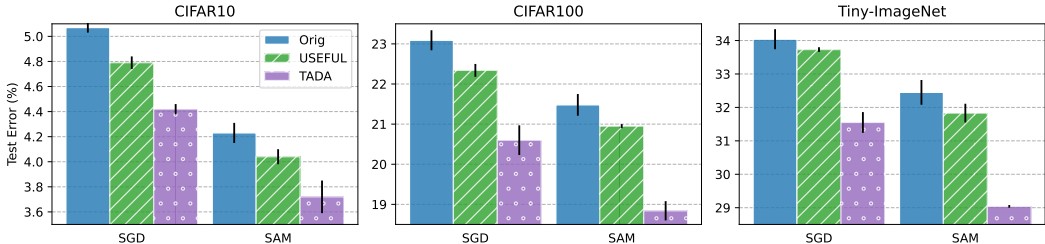

Figure 2: Test classification error of ResNet18 on CIFAR10, CIFAR100 and TinyImageNet. For USEFUL, we use a factor of $k = 2$, as higher $k$ harms the performance. In contrast for TADA, $k = 5, 5, 4$ for CIFAR10, CIFAR100, and Tiny-ImageNet, respectively. TADA improves both SGD and SAM. Notably, it enables SGD to outperform SAM on CIFAR100 and TinyImageNet.

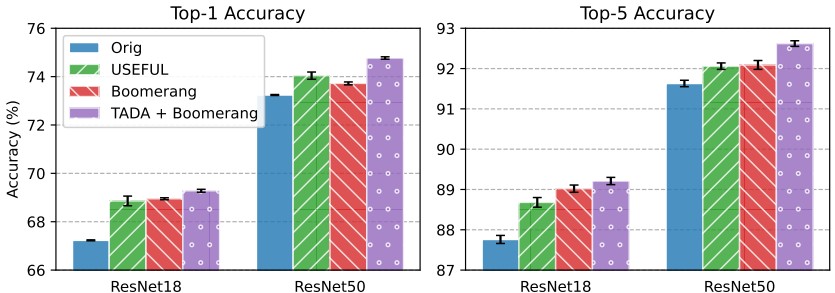

Figure 3: Top-1 and top-5 accuracies of training ResNet18 and ResNet50 on ImageNet.

et al., 2009), Flowers-102 (Nilsback & Zisserman, 2008), Aircraft (Maji et al., 2013), and Stanford Cars (Krause et al., 2013).

**Augmented training datasets.** We train different models on: (1) **Original**: The original training datasets without any modifications. (2) **USEFUL** (Nguyen et al., 2024): Augmented dataset with upsampled real images that are not learned in early training. (3) **TADA**: Replace the upsampled samples in USEFUL with their corresponding synthetic images. For $k > 2$, we use generated images at different diffusion denoising steps, instead of generating from scratch.

**Hyperparameters.** For USEFUL, we set the upsampling factor $k$ to 2 as it yields the best performance. For TADA, we use $k = 5$ for CIFAR10 and CIFAR100 and $k = 4$ for TinyImageNet. For generating synthetic images using GLIDE, we use guidance scale of 3 and run denoising for 100 steps, saving generated images every 10 steps. More training details are in Appendix F.1.

### 5.1 TADA IS EFFECTIVE ACROSS DATASETS AND ARCHITECTURES

**Different datasets.** Figure 2 clearly shows that TADA significantly reduces the test classification error compared to both the **Original** and **USEFUL** methods across all datasets, namely CIFAR10, CIFAR100, and Tiny-ImageNet. For Tiny ImageNet, TADA yields an improvement of 2.8% when training with SAM. The superior performance of TADA compared to USEFUL is well aligned with our theoretical results in Section 4.3. Notably, SGD with TADA outperforms SAM on CIFAR100 and TinyImageNet, further confirming the effectiveness of our approach. Importantly, this advantage extends to large-scale settings in Figure 3. TADA achieves the highest Top-1 and Top-5 accuracies on ImageNet when training both ResNet18 and ResNet50. Moreover, it outperforms Boomerang while augmenting only 65% of the dataset, compared to Boomerang's 100% synthetic augmentation.

**Different model architectures.** To further evaluate the generalization of our approach, we conduct experiments on multiple model architectures using **CIFAR10** as the base dataset. Specifically, we apply TADA to CNNs (VGG19, DenseNet121) and Transformers (ViT-S). Figure 4 presents the test classification error for different architectures. The results demonstrate that TADA achieves consistently lower classification error than both the **Original** and **USEFUL** methods across all architectures, under both SGD and SAM optimization settings. Moreover, when applied to

Figure 4: Test classification error of VGG19, DenseNet121, and ViT-S on CIFAR10. For USEFUL, we use a factor of $k = 2$—as higher $k$ hurst the performance—while for TADA, we use $k = 5$.

Table 1: Test error of ConvNeXt-T and Swin-T on CIFAR-10 using SGD and $k = 2$.

| Method | ConvNeXtT | SwinT |
|---|---|---|
| Original | $37.33 \pm 3.12$ | $16.10 \pm 0.19$ |
| USEFUL | $34.16 \pm 2.47$ | $14.93 \pm 0.07$ |
| TADA | $\mathbf{27.40 \pm 1.99}$ | $\mathbf{14.57 \pm 0.10}$ |

Table 2: Test error of pre-trained ResNet18 on Flowers-102, Aircraft, and Stanford Cars datasets.

| Method | Flowers-102 | Aircraft | Stanford Cars |
|---|---|---|---|
| Original | $8.55 \pm 0.19$ | $26.02 \pm 0.27$ | $15.45 \pm 0.02$ |
| DiffuseMix | $8.92 \pm 0.15$ | $25.65 \pm 0.20$ | $15.19 \pm 0.06$ |
| TADA+DifMx | $\mathbf{8.08 \pm 0.16}$ | $\mathbf{25.12 \pm 0.35}$ | $\mathbf{14.96 \pm 0.07}$ |

Figure 5: (left & middle) Comparison between different synthetic image augmentation strategies when training ResNet18 on CIFAR10 and CIFAR100. For Syn-rand and TADA, we use $k = 2$ resulting in only 30% and 40% additional examples compared to 100% of Syn-all. (right) TADA with $k = 5$ can be stacked with TrivialAugment (TA) to further boosts the performance when training ResNet18 on CIFAR10, achieving (to our knowledge) SOTA test classification error.

state-of-the-art architectures such as ConvNeXt (Liu et al., 2022) and Swin Transformer (Liu et al., 2021), TADA still outperforms the baselines by a substantial margin, as reported in Table 1. These findings confirm the effectiveness of our approach across different model structures.

**Transfer learning and stacking with other synthetic image generations.** We evaluate TADA in a transfer learning setting, where we fine-tune a ResNet18 pre-trained on ImageNet-1K on 3 popular fine-grained image classification datasets including Flowers-102, Aircraft, and Stanford Cars datasets. Table 2 compares TADA with DiffuseMix (Islam et al., 2024a) which is a SOTA data augmentation method. It enhances diversity of synthetic images by blending partial natural images with images generated via InstructPix2Pix (Brooks et al., 2023) diffusion model. Applying TADA to augment slow-learnable images with DiffuseMix significantly outperform no-augmentation and augmenting entire dataset with DiffuseMix.

**Do We Need All the Synthetic Data?** To answer this question, we compare synthetically augmenting all examples (**Syn-all**)–which doubles the training set size–with TADA with $k = 2$–which results in increase of approximately 30% and 40% of the total training data–in CIFAR10 and CIFAR100. Notably, the generation time for TADA is reduced to 0.3x and 0.4x that of **Syn-all**, making it more efficient. In addition, we consider a baseline (**Syn-rand**) where random images are augmented with their corresponding synthetic ones. Figure 5 shows that TADA has a much lower test classification error compared to **Syn-rand** (same cost) and **Syn-all** (higher cost). This highlights the effectiveness of our *targeted* data augmentation.

## 5.2 ABLATION STUDIES

**Application to non-diffusion augmentation.** Beyond diffusion-based augmentation, our targeted augmentation generalizes to traditional data augmentation. We evaluate this by applying TrivialAugment (TA) (Müller & Hutter, 2021) to augment upsampled data. As summarized in Table 5, targeted augmentation consistently outperforms full-data augmentation.

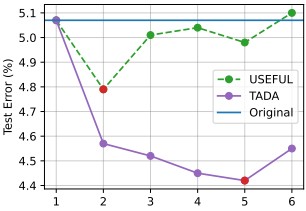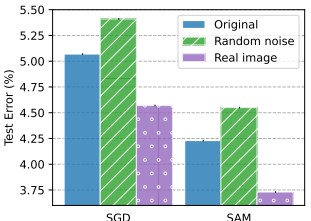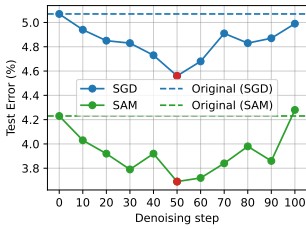

Figure 6: Training ResNet18 on CIFAR10. (left) The effect of amplification factor $k$ on test error for upsampling vs generation. Red points indicate the optimal choice of $k$. $k > 2$ hurts upsampling but boosts generation. (middle) Generating synthetic CIFAR10 images from real images outperform starting from random noise. (right) Effect of the number of denoising steps on the performance with $k = 2$.

**TADA stacked with strong augmentation.** Figure 5 right shows that TADA stacked with Triv-ialAugment achieves state-of-the-art results when training ResNet18 on CIFAR10. Appendix F.2 shows similar results for CIFAR100 and Tiny-ImageNet.

**Identifying slow-learnable features.** Table 8 in Appendix F.2 shows that identifying slow-learnable features by clustering model outputs outperforms selection based on high-loss or misclassification.

**For larger $k$, upsampling hurts but generation helps.** Figure 6a illustrates the performance of TADA and USEFUL on CIFAR10 when varying the upsampling factors. Upsampling achieves the best performance at $k = 2$ due to overfitting noise at larger $k$. But using synthetic images, TADA benefits from larger values of $k$, yielding the best performance at $k = 5$ for CIFAR10 and CIFAR100, and $k = 4$ for Tiny-ImageNet. Detailed results for CIFAR100 and Tiny-ImageNet can be found in Table 6 in Appendix F.2. This corroborates our theoretical findings in Section 4.2.

**Choices of initialization for denoising.** We compare generating synthetic images by adding noise and denoising real images with denoising from random noises. Figure 6b illustrates that real data guidance is necessary for targeted synthetic image augmentation. When using random noises to generate synthetic images, performance of ResNet18 on the augmented training datasets is even worse than that on the original set, as generated images do not effectively amplify slow-learnable features.

**Number of denoising steps.** Figure 6c demonstrates the effect of the number of denoising steps on the performance of TADA. Using fewer steps generates images that are too close to real images, amplifying the noise in the real images. In contrast, using too many steps results in too much noise, which also harms the performance. Overall, using 50 steps yields the best results for both SGD and SAM.

**Convergence.** Figure 8 in Appendix F.2 shows the mini-batch gradient variance when training ResNet18 on the augmented CIFAR10 dataset using upsampling and generation. Generation results in lower gradient variance compared to upsampling for different $k$, confirming our results in Sec. 4.3.

**Object detection experiments.** On MS-COCO object detection (Table 10), TADA consistently improves both AP50 and mAP50 $-$ 95 when training YOLOv5m from scratch. It outperforms In-stanceAugmentation and all other baselines while using 25% fewer augmented images. These results demonstrate that TADA extends beyond classification and remains effective for dense prediction tasks.

## 6  CONCLUSION

Diffusion-based synthetic data augmentation improves generalization but often enlarges datasets by 10–30×, incurring high computational cost and limited diversity control. We proposed TADA (**TA**rgeted **D**iffusion **A**ugmentation), a principled framework that selectively augments examples not learned early in training using faithful synthetic images with diverse noise. We showed that augmenting only this targeted subset consistently outperforms full augmentation, and our theoretical analysis explains the gains through more uniform feature learning without noise amplification. Across architectures (ResNet, ViT, ConvNeXt, Swin), datasets (CIFAR-10/100, TinyImageNet, ImageNet), and optimizers (SGD, SAM), augmenting just 30–40% of the data yields up to 2.8% improvement. Notably, TADA with SGD surpasses SAM on CIFAR-100 and TinyImageNet. Results on object detection further demonstrate its effectiveness beyond classification.

## ACKNOWLEDGEMENTS

This research was supported in part by in part by the NSF CAREER Award 2146492, NSF-Simons AI Institute for Cosmic Origins (CosmicAI), and NSF AI Institute for Foundations of Machine Learning (IFML).

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

# A   FORMAL PROOFS

## A.1   FULL GRADIENT FORMULAS & USEFUL LEMMAS

From the setting, at iteration $t$, we can take the derivative with respect to the j-th filter $\boldsymbol{w}_j^{(t)}$ in the following three training schemes:

1. GD training on the augmented dataset via upsampling: Lemmas A.1, A.2.
2. GD training on the augmented dataset via synthetic generation: Lemmas A.3, A.4.
3. SAM and GD training on the original dataset: Lemmas A.5, A.6.

Recall that $k$ is the augmenting factor, and the augmented datasets $D_G$, $D_U$ now have size $N_{\text{new}} = \alpha N + k(1 - \alpha)N$.

**Lemma A.1.** *(Upsampling: full gradient) In the augmented dataset $D_U$ after upsampling the "slow-learnable" subset with a factor $k$, for $t \geq 0$ and $j \in [J]$, the gradient of the loss $\mathcal{L}^U(\boldsymbol{W}^{(t)})$ with respect to $\boldsymbol{w}_j^{(t)}$ is*

$$\nabla_{\boldsymbol{w}_j^{(t)}}^U \mathcal{L}(\boldsymbol{W}^{(t)}) = -\frac{3}{N_{new}} \sum_{i=1}^{\alpha N} l_i^{(t)} \left( \beta_e^3 \langle \boldsymbol{w}_j^{(t)}, \boldsymbol{v}_e \rangle^2 \boldsymbol{v}_e + y_i \langle \boldsymbol{w}_j^{(t)}, \boldsymbol{\xi}_i \rangle^2 \boldsymbol{\xi}_i \right)$$

$$- \frac{3k}{N_{new}} \sum_{i=\alpha N+1}^{N} k l_i^{(t)} \left( \beta_d^3 \langle \boldsymbol{w}_j^{(t)}, \boldsymbol{v}_d \rangle^2 \boldsymbol{v}_d + y_i \langle \boldsymbol{w}_j^{(t)}, \boldsymbol{\xi}_i \rangle^2 \boldsymbol{\xi}_i \right),$$

*where $l_i^{(t)} = sigmoid(-y_i f(\boldsymbol{x}_i; \boldsymbol{W}^{(t)}))$ for the two-layer CNN model $f$.*

*Proof.* We compute the gradient directly from the loss function:

$$\nabla_{\boldsymbol{w}_j^{(t)}}^U \mathcal{L}(\boldsymbol{W}^{(t)}) = -\frac{1}{N_{\text{new}}} \sum_{i=1}^{N_{\text{new}}} \frac{\exp(-y_i f(\boldsymbol{x}_i; \boldsymbol{W}^{(t)}))}{1 + \exp(-y_i f(\boldsymbol{x}_i; \boldsymbol{W}^{(t)}))} y_i \nabla_{\boldsymbol{w}_j^{(t)}} f(\boldsymbol{x}_i; \boldsymbol{W}^{(t)})$$

$$= -\frac{3}{N_{\text{new}}} \sum_{i=1}^{N_{\text{new}}} l_i^{(t)} y_i \sum_{p=1}^{P} \langle \boldsymbol{w}_j^{(t)}, \boldsymbol{x}^{(p)} \rangle^3$$

$$= -\frac{3}{N_{\text{new}}} \sum_{i=1}^{\alpha N} l_i^{(t)} \left( \beta_e^3 \langle \boldsymbol{w}_j^{(t)}, \boldsymbol{v}_e \rangle^2 \boldsymbol{v}_e + y_i \langle \boldsymbol{w}_j^{(t)}, \boldsymbol{\xi}_i \rangle^2 \boldsymbol{\xi}_i \right)$$

$$- \frac{3}{N_{\text{new}}} \sum_{i=\alpha N+1}^{N_{\text{new}}} l_i^{(t)} \left( \beta_d^3 \langle \boldsymbol{w}_j^{(t)}, \boldsymbol{v}_d \rangle^2 \boldsymbol{v}_d + y_i \langle \boldsymbol{w}_j^{(t)}, \boldsymbol{\xi}_i \rangle^2 \boldsymbol{\xi}_i \right)$$

$$= -\frac{3}{N_{\text{new}}} \sum_{i=1}^{\alpha N} l_i^{(t)} \left( \beta_e^3 \langle \boldsymbol{w}_j^{(t)}, \boldsymbol{v}_e \rangle^2 \boldsymbol{v}_e + y_i \langle \boldsymbol{w}_j^{(t)}, \boldsymbol{\xi}_i \rangle^2 \boldsymbol{\xi}_i \right)$$

$$- \frac{3k}{N_{\text{new}}} \sum_{i=\alpha N+1}^{N} l_i^{(t)} \left( \beta_d^3 \langle \boldsymbol{w}_j^{(t)}, \boldsymbol{v}_d \rangle^2 \boldsymbol{v}_d + y_i \langle \boldsymbol{w}_j^{(t)}, \boldsymbol{\xi}_i \rangle^2 \boldsymbol{\xi}_i \right),$$

where the last step follows from the fact that there are $k$ copies of the same subset in the upsampled portion of the data.                   $\square$

**Lemma A.2.** *(Upsampling: feature & noise gradients) In the same setting, $\nabla_{\boldsymbol{w}_j^{(t)}}^U \mathcal{L}(\boldsymbol{W}^{(t)})$ learns the features and noises as follows:*

1. *Fast-learnable feature gradient:*

$$\langle \nabla_{\boldsymbol{w}_j^{(t)}}^U \mathcal{L}(\boldsymbol{W}^{(t)}), \boldsymbol{v}_e \rangle = -\frac{3\beta_e^3}{N_{new}} \sum_{i=1}^{\alpha N} l_i^{(t)} \langle \boldsymbol{w}_j^{(t)}, \boldsymbol{v}_e \rangle^2$$

2. *Slow-learnable feature gradient*:

$$\langle \nabla^U_{\boldsymbol{w}_j^{(t)}} \mathcal{L}(\boldsymbol{W}^{(t)}), \boldsymbol{v}_d \rangle = -\frac{3k\beta_d^3}{N_{new}} \sum_{i=\alpha N+1}^{N} l_i^{(t)} \langle \boldsymbol{w}_j^{(t)}, \boldsymbol{v}_d \rangle^2$$

3. *Noise gradient*:

*(a). For $\boldsymbol{\xi}_i$, $i = 1, \cdots, \alpha N$*

$$\langle \nabla^U_{\boldsymbol{w}_j^{(t)}} \mathcal{L}(\boldsymbol{W}^{(t)}), \boldsymbol{\xi}_i \rangle = -\frac{3}{N_{new}} l_i^{(t)} y_i \langle \boldsymbol{w}_j^{(t)}, \boldsymbol{\xi}_i \rangle^2 \|\boldsymbol{\xi}_i\|^2.$$

*(b). For $\boldsymbol{\xi}_i$, $i = \alpha N + 1, \cdots, N$*

$$\langle \nabla^U_{\boldsymbol{w}_j^{(t)}} \mathcal{L}(\boldsymbol{W}^{(t)}), \boldsymbol{\xi}_i \rangle = -\frac{3k}{N_{new}} l_i^{(t)} y_i \langle \boldsymbol{w}_j^{(t)}, \boldsymbol{\xi}_i \rangle^2 \|\boldsymbol{\xi}_i\|^2$$

*Proof.* The proof follows directly from taking the inner product with the gradient formula in Lemma A.1. Then we proceed using that $\boldsymbol{v}_e, \boldsymbol{v}_d, \boldsymbol{\xi}_i$ form a orthogonal set and that summations involving noise cross-terms become negligible. $\square$

**Lemma A.3.** *(Generation: full gradient) In the augmented dataset $D_G$ after synthetically generating the "slow-learnable" subset with a factor $k$, for $t \geq 0$ and $j \in [J]$, the gradient of the loss $\mathcal{L}^G(\boldsymbol{W}^{(t)})$ with respect to $\boldsymbol{w}_j^{(t)}$ is*

$$\nabla^U_{\boldsymbol{w}_j^{(t)}} \mathcal{L}(\boldsymbol{W}^{(t)}) = -\frac{3}{N_{new}} \sum_{i=1}^{\alpha N} l_i^{(t)} \left( \beta_e^3 \langle \boldsymbol{w}_j^{(t)}, \boldsymbol{v}_e \rangle^2 \boldsymbol{v}_e + y_i \langle \boldsymbol{w}_j^{(t)}, \boldsymbol{\xi}_i \rangle^2 \boldsymbol{\xi}_i \right)$$

$$-\frac{3}{N_{new}} \sum_{i=\alpha N+1}^{N} l_i^{(t)} \left( \beta_d^3 \langle \boldsymbol{w}_j^{(t)}, \boldsymbol{v}_d \rangle^2 \boldsymbol{v}_d + y_i \langle \boldsymbol{w}_j^{(t)}, \boldsymbol{\xi}_i \rangle^2 \boldsymbol{\xi}_i \right)$$

$$-\frac{3}{N_{new}} \sum_{i=N+1}^{N_{new}} l_i^{(t)} \left( \beta_d^3 \langle \boldsymbol{w}_j^{(t)}, \boldsymbol{v}_d \rangle^2 \boldsymbol{v}_d + y_i \langle \boldsymbol{w}_j^{(t)}, \boldsymbol{\gamma}_i \rangle^2 \boldsymbol{\gamma}_i \right),$$

*where $l_i^{(t)} = sigmoid(-y_i f(\boldsymbol{x}_i; \boldsymbol{W}^{(t)}))$ for the two-layer CNN model $f$.*

*Proof.* The proof is similar to that of Lemma A.1, except that for data that contain $\boldsymbol{v}_d$, the first $(1 - \alpha)N$ data points come from the original dataset, and the rest comes from synthetic generation with noise $\boldsymbol{\gamma}_i$, $i = N + 1, \ldots, N_{\text{new}}$. $\square$

**Lemma A.4.** *(Generation: feature & noise gradients) In the same setting, $\nabla^G_{\boldsymbol{w}_j^{(t)}} \mathcal{L}(\boldsymbol{W}^{(t)})$ learns the features and noises as follows:*

1. *Fast-learnable feature gradient*:

$$\langle \nabla^G_{\boldsymbol{w}_j^{(t)}} \mathcal{L}(\boldsymbol{W}^{(t)}), \boldsymbol{v}_e \rangle = -\frac{3\beta_e^3}{N_{new}} \sum_{i=1}^{\alpha N} l_i^{(t)} \langle \boldsymbol{w}_j^{(t)}, \boldsymbol{v}_e \rangle^2$$

2. *Slow-learnable feature gradient*:

$$\langle \nabla^G_{\boldsymbol{w}_j^{(t)}} \mathcal{L}(\boldsymbol{W}^{(t)}), \boldsymbol{v}_d \rangle = -\frac{3\beta_d^3}{N_{new}} \sum_{i=\alpha N+1}^{N_{new}} l_i^{(t)} \langle \boldsymbol{w}_j^{(t)}, \boldsymbol{v}_d \rangle^2$$

3. *Noise gradient: Let $\{\boldsymbol{\phi}_i\}_{i=1}^{N_{new}} = \{\boldsymbol{\xi}_i\}_{i=1}^{N} \cup \{\boldsymbol{\gamma}_i\}_{i=N+1}^{N_{new}}$ denote the set of noises in $D_G$ (which can be the original or generation noise). Then for any $\boldsymbol{\phi}_i$, $i \in [N_{new}]$,*

$$\langle \nabla^G_{\boldsymbol{w}_j^{(t)}} \mathcal{L}(\boldsymbol{W}^{(t)}), \boldsymbol{\phi}_i \rangle = -\frac{3}{N_{new}} l_i^{(t)} y_i \langle \boldsymbol{w}_j^{(t)}, \boldsymbol{\phi}_i \rangle^2 \|\boldsymbol{\phi}_i\|^2.$$

*Proof.* Similar to Lemma A.2. We directly take the inner product and assume that all the noises are dispersed such that summations involving their cross-terms become insignificant in high dimension. Recall we also assume that the generation noise is orthogonal to features and independent of $\boldsymbol{\xi}_i$'s. $\quad\square$

Following the same process, we have the following gradients for SAM and GD on the original dataset.

**Lemma A.5.** *(Original dataset: SAM & GD gradients) In the original dataset $D$, for $t \geq 0$ and $j \in [J]$ of GD, the gradient of the loss $\mathcal{L}(\boldsymbol{W}^{(t)})$ with respect to $\boldsymbol{w}_j^{(t)}$ is*

$$\nabla_{\boldsymbol{w}_j^{(t)}} \mathcal{L}(\boldsymbol{W}^{(t)}) = -\frac{3}{N} \sum_{i=1}^{\alpha N} l_i^{(t)} \left( \beta_e^3 \langle \boldsymbol{w}_j^{(t)}, \boldsymbol{v}_e \rangle^2 \boldsymbol{v}_e + y_i \langle \boldsymbol{w}_j^{(t)}, \boldsymbol{\xi}_i \rangle^2 \boldsymbol{\xi}_i \right)$$

$$-\frac{3}{N} \sum_{i=\alpha N+1}^{N} l_i^{(t)} \left( \beta_d^3 \langle \boldsymbol{w}_j^{(t)}, \boldsymbol{v}_d \rangle^2 \boldsymbol{v}_d + y_i \langle \boldsymbol{w}_j^{(t)}, \boldsymbol{\xi}_i \rangle^2 \boldsymbol{\xi}_i \right),$$

*where $l_i^{(t)} = sigmoid(-y_i f(\boldsymbol{x}_i; \boldsymbol{W}^{(t)}))$ for the two-layer CNN model $f$.*

*Suppose we train with SAM. Then the perturbed gradient has the same expression except that we replace $\boldsymbol{w}_j^{(t)}$ with the perturbed weights $\boldsymbol{w}_{j,\boldsymbol{\epsilon}}^{(t)}$ and replace $l_i^{(t)}$ with $l_{i,\boldsymbol{\epsilon}}^{(t)} = sigmoid(-y_i f(\boldsymbol{x}_i; \boldsymbol{W}^{(t)} + \boldsymbol{\epsilon}^{(t)}))$:*

$$\nabla_{\boldsymbol{w}_{j,\boldsymbol{\epsilon}}^{(t)}} \mathcal{L}(\boldsymbol{W}^{(t)} + \boldsymbol{\epsilon}^{(t)}) = \nabla_{\boldsymbol{w}_{j,\boldsymbol{\epsilon}}^{(t)}} \mathcal{L} \left( \boldsymbol{W}^{(t)} + \frac{\rho}{\|\nabla(\mathcal{L}(\boldsymbol{W}^{(t)}))\|_F} \nabla(\mathcal{L}(\boldsymbol{W}^{(t)})) \right)$$

$$= -\frac{3}{N} \sum_{i=1}^{\alpha N} l_{i,\boldsymbol{\epsilon}}^{(t)} \left( \beta_e^3 \langle \boldsymbol{w}_{j,\boldsymbol{\epsilon}}^{(t)}, \boldsymbol{v}_e \rangle^2 \boldsymbol{v}_e + y_i \langle \boldsymbol{w}_{j,\boldsymbol{\epsilon}}^{(t)}, \boldsymbol{\xi}_i \rangle^2 \boldsymbol{\xi}_i \right)$$

$$-\frac{3}{N} \sum_{i=\alpha N+1}^{N} l_{i,\boldsymbol{\epsilon}}^{(t)} \left( \beta_d^3 \langle \boldsymbol{w}_{j,\boldsymbol{\epsilon}}^{(t)}, \boldsymbol{v}_d \rangle^2 \boldsymbol{v}_d + y_i \langle \boldsymbol{w}_{j,\boldsymbol{\epsilon}}^{(t)}, \boldsymbol{\xi}_i \rangle^2 \boldsymbol{\xi}_i \right).$$

**Lemma A.6.** *(Original dataset: SAM & GD feature & noise gradients) In the same setting, $\nabla_{\boldsymbol{w}_j^{(t)}} \mathcal{L}(\boldsymbol{W}^{(t)})$ learns the features and noises as follows:*

1. *Fast-learnable feature gradient:*

$$\langle \nabla_{\boldsymbol{w}_j^{(t)}} \mathcal{L}(\boldsymbol{W}^{(t)}), \boldsymbol{v}_e \rangle = -\frac{3\beta_e^3}{N} \sum_{i=1}^{\alpha N} l_i^{(t)} \langle \boldsymbol{w}_j^{(t)}, \boldsymbol{v}_e \rangle^2$$

2. *Slow-learnable feature gradient:*

$$\langle \nabla_{\boldsymbol{w}_j^{(t)}} \mathcal{L}(\boldsymbol{W}^{(t)}), \boldsymbol{v}_d \rangle = -\frac{3\beta_d^3}{N} \sum_{i=\alpha N+1}^{N} l_i^{(t)} \langle \boldsymbol{w}_j^{(t)}, \boldsymbol{v}_d \rangle^2$$

3. *Noise gradient:*

$$\langle \nabla_{\boldsymbol{w}_j^{(t)}} \mathcal{L}(\boldsymbol{W}^{(t)}), \boldsymbol{\xi}_i \rangle = -\frac{3}{N} l_i^{(t)} y_i \langle \boldsymbol{w}_j^{(t)}, \boldsymbol{\xi}_i \rangle^2 \|\boldsymbol{\xi}_i\|^2.$$

*The SAM feature & noise gradients are similar except that we replace $\boldsymbol{w}_j^{(t)}, l_i^{(t)}$ with $\boldsymbol{w}_{j,\boldsymbol{\epsilon}}^{(t)}, l_{i,\boldsymbol{\epsilon}}^{(t)}$.*

**Remark.** Suppose the data point $\boldsymbol{x}_i$ has feature $\boldsymbol{v}_d$ and noise $\boldsymbol{\phi}_i$. We have that

$$l_i^{(t)} = \text{sigmoid} \left( \sum_{j=1}^{J} -\beta_d^3 \langle \boldsymbol{w}_j^{(t)}, \boldsymbol{v}_d \rangle^3 - y_i \langle \boldsymbol{w}_j^{(t)}, \boldsymbol{\phi}_i \rangle^3 \right), \tag{3}$$

$$\text{and } l_{i,\boldsymbol{\epsilon}}^{(t)} = \text{sigmoid}\left(\sum_{j=1}^{J} -\beta_d^3 \langle \boldsymbol{w}_{j,\boldsymbol{\epsilon}}^{(t)}, \boldsymbol{v}_d \rangle^3 - y_i \langle \boldsymbol{w}_{j,\boldsymbol{\epsilon}}^{(t)}, \boldsymbol{\phi}_i \rangle^3 \right).$$

The same formula holds if the feature is $\boldsymbol{v}_e$.

Similar to Nguyen et al. (2024), we assume that the logit terms are controlled in some sense to reduce the nonlinearity. More formally, for Theorem 4.1, we will use the approximation $l_i^{(t)} = l_{i,\boldsymbol{\epsilon}}^{(t)} = \Theta(1)$ for any early iteration $t$. At a high level, this follows from the small perturbation and the fact that the weights have not "significantly" learned any noises (despite some overfitting) in the early phase.

$$\text{sigmoid}\left(\sum_{j=1}^{J} -\beta_d^3 \langle \boldsymbol{w}_j^{(t)}, \boldsymbol{v}_d \rangle^3 - y_i \langle \boldsymbol{w}_j^{(t)}, \boldsymbol{\phi}_i \rangle^3 \right) = \text{sigmoid}\left(\sum_{j=1}^{J} -\beta_d^3 \langle \boldsymbol{w}_{j,\boldsymbol{\epsilon}}^{(t)}, \boldsymbol{v}_d \rangle^3 - y_i \langle \boldsymbol{w}_{j,\boldsymbol{\epsilon}}^{(t)}, \boldsymbol{\phi}_i \rangle^3 \right).$$

Hence, this is implicitly assuming a small $\rho$ such that $\boldsymbol{w}_j^{(t)}$ and $\boldsymbol{w}_{j,\boldsymbol{\epsilon}}^{(t)}$ are not too different, and a large number of neurons $J$ such that the noise term in the sigmoid $\sum_{j=1}^{J} y_i \langle \boldsymbol{w}_{j,\boldsymbol{\epsilon}}^{(t)}, \boldsymbol{\phi}_i \rangle^3$ is 'more 'diluted" (close to 0).

**Insights from gradient formulas.** By directly computing these relevant gradients, we can see that some observed phenomena already become self-explanatory in the formulas. For instance, Lemma A.2 implies that for upsampling, the gradient alignment is $k$ times larger for the noises we replicated. At a high level, this would cause overfitting to these particular noises when performing gradient update. In the next section, we formalize the intuition by examining the underlying mechanism of noise learning in greater detail.

## B  GD VS. SAM NOISE LEARNING: PROOF OF THEOREM 4.1

**Remark.** Heuristically, we say that the noise is being learned well if the magnitude of the alignment $|\langle \boldsymbol{w}_j^{(t)}, \boldsymbol{\xi}_i \rangle|$ is large, meaning that the weights align or misalign with the noise to a great extent. We say that the model is learning the noise if $|\langle \boldsymbol{w}_j^{(t)}, \boldsymbol{\xi}_i \rangle|$ increases over time. In practice, overly fitting or overly avoiding certain noises are both harmful to a model's generalization.

**Lemma B.1.** *(Gradient norm bound) In our setting, we can bound the norm of the gradient matrix* $\nabla \mathcal{L}(\boldsymbol{W}^{(t)})$ *as:*

$$\|\nabla \mathcal{L}(\boldsymbol{W}^{(t)})\|_F \geq \frac{3}{N} l_i^{(t)} \langle \boldsymbol{w}_j^{(t)}, \boldsymbol{\xi}_i \rangle^2 \|\boldsymbol{\xi}_i\| \quad \forall i.$$

*Proof.* This norm can be lower bounded by the the norm of one column:

$$\|\nabla \mathcal{L}(\boldsymbol{W}^{(t)})\|_F = \sqrt{\sum_{j=1}^{J} \left\| \nabla_{\boldsymbol{w}_j^{(t)}} \mathcal{L}(\boldsymbol{W}^{(t)}) \right\|^2}$$

$$\geq \left\| \nabla_{\boldsymbol{w}_j^{(t)}} \mathcal{L}(\boldsymbol{W}^{(t)}) \right\| \quad \text{for some neuron } j$$

$$= \sqrt{\frac{9\beta_e^6}{N^2} \langle \boldsymbol{w}_j^{(t)}, \boldsymbol{v}_e \rangle^4 \left(\sum_{i=1}^{\alpha N} l_i^{(t)}\right)^2 + \frac{9\beta_d^6}{N^2} \langle \boldsymbol{w}_j^{(t)}, \boldsymbol{v}_d \rangle^4 \left(\sum_{i=\alpha N+1}^{N} l_i^{(t)}\right)^2 + \left\| \frac{3}{N} \sum_{i=1}^{N} l_i^{(t)} y_i \langle \boldsymbol{w}_j^{(t)}, \boldsymbol{\xi}_i \rangle^2 \boldsymbol{\xi}_i \right\|^2}$$

$$\text{by taking the norm of the gradient in Lemma A.5 and using orthogonality}$$

$$\geq \left\| \frac{3}{N} \sum_{i=1}^{N} l_i^{(t)} y_i \langle \boldsymbol{w}_j^{(t)}, \boldsymbol{\xi}_i \rangle^2 \boldsymbol{\xi}_i \right\| \geq \left\| \frac{3}{N} l_i^{(t)} y_i \langle \boldsymbol{w}_j^{(t)}, \boldsymbol{\xi}_i \rangle^2 \boldsymbol{\xi}_i \right\| = \frac{3}{N} l_i^{(t)} \langle \boldsymbol{w}_j^{(t)}, \boldsymbol{\xi}_i \rangle^2 \|\boldsymbol{\xi}_i\| \quad \forall i$$

$$\text{since cross terms become negligible}$$

$\square$

**Lemma B.2.** *(Noise set equivalences) With a sufficiently small SAM perturbation parameter $\rho$, at some early iteration $t$, the following set relations hold:*

$$\mathcal{I}_{j,+}^{(t)} = \mathcal{I}_{j,\boldsymbol{\epsilon},+}^{(t)}, \quad \mathcal{I}_{j,-}^{(t)} = \mathcal{I}_{j,\boldsymbol{\epsilon},-}^{(t)}.$$

*Proof.* We first note that for some early training weights $\boldsymbol{W}^{(t)}$, we have the following connection between the original weights and the SAM perturbed weights:

$$
\begin{aligned}
\langle \boldsymbol{w}_{j,\boldsymbol{\epsilon}}^{(t)}, \boldsymbol{\xi}_i \rangle &= \langle \boldsymbol{w}_j^{(t)} + \rho^{(t)} \nabla_{\boldsymbol{w}_j^{(t)}} \mathcal{L}(\boldsymbol{W}^{(t)}), \boldsymbol{\xi}_i \rangle \\
&= \langle \boldsymbol{w}_j^{(t)}, \boldsymbol{\xi}_i \rangle + \rho^{(t)} \langle \nabla_{\boldsymbol{w}_j^{(t)}} \mathcal{L}(\boldsymbol{W}^{(t)}), \boldsymbol{\xi}_i \rangle \\
&= \langle \boldsymbol{w}_j^{(t)}, \boldsymbol{\xi}_i \rangle - \frac{3\rho^{(t)}}{N} l_i^{(t)} y_i \langle \boldsymbol{w}_j^{(t)}, \boldsymbol{\xi}_i \rangle^2 \|\boldsymbol{\xi}_i\|^2 \quad \text{by Lemma A.6} \quad (4)
\end{aligned}
$$

We consider sufficiently small $\rho$ such that:

$$
0 < \rho < \min \left\{ \frac{|\langle \boldsymbol{w}_j^{(t)}, \boldsymbol{\xi}_i \rangle|}{\|\boldsymbol{\xi}_i\|} : \boldsymbol{\xi}_i \in D \right\}. \quad (5)
$$

We recall that $\rho^{(t)} = \rho/\|\nabla \mathcal{L}(\boldsymbol{W}^{(t)})\|_F$. Then suppose $\boldsymbol{\xi}_i \in \mathcal{I}_{j,-}^{(t)}$, that is $y_i$ and $\langle \boldsymbol{w}_j^{(t)}, \boldsymbol{\xi}_i \rangle$ have opposite signs. Eq. 4 implies

$$
\begin{aligned}
\langle \boldsymbol{w}_{j,\boldsymbol{\epsilon}}^{(t)}, \boldsymbol{\xi}_i \rangle &= \langle \boldsymbol{w}_j^{(t)}, \boldsymbol{\xi}_i \rangle \left( 1 - \frac{3\rho^{(t)}}{N} l_i^{(t)} y_i \langle \boldsymbol{w}_j^{(t)}, \boldsymbol{\xi}_i \rangle \|\boldsymbol{\xi}_i\|^2 \right) \\
&= \langle \boldsymbol{w}_j^{(t)}, \boldsymbol{\xi}_i \rangle \underbrace{\left( 1 + \frac{3\rho^{(t)}}{N} l_i^{(t)} \|\boldsymbol{\xi}_i\|^2 |\langle \boldsymbol{w}_j^{(t)}, \boldsymbol{\xi}_i \rangle| \right)}_{>0},
\end{aligned}
$$

which implies that $\operatorname{sgn}(\langle \boldsymbol{w}_{j,\boldsymbol{\epsilon}}^{(t)}, \boldsymbol{\xi}_i \rangle) = \operatorname{sgn}(\langle \boldsymbol{w}_j^{(t)}, \boldsymbol{\xi}_i \rangle)$ and $\boldsymbol{\xi}_i \in \mathcal{I}_{j,\boldsymbol{\epsilon},-}^{(t)}$.

Now suppose $\boldsymbol{\xi}_i \in \mathcal{I}_{j,+}^{(t)}$. Eq. 4 becomes:

$$
\begin{aligned}
\langle \boldsymbol{w}_{j,\boldsymbol{\epsilon}}^{(t)}, \boldsymbol{\xi}_i \rangle &= \langle \boldsymbol{w}_j^{(t)}, \boldsymbol{\xi}_i \rangle \left( 1 - \frac{3\rho^{(t)}}{N} l_i^{(t)} y_i \langle \boldsymbol{w}_j^{(t)}, \boldsymbol{\xi}_i \rangle \|\boldsymbol{\xi}_i\|^2 \right) \\
&= \langle \boldsymbol{w}_j^{(t)}, \boldsymbol{\xi}_i \rangle \underbrace{\left( 1 - \frac{3\rho^{(t)}}{N} l_i^{(t)} \|\boldsymbol{\xi}_i\|^2 |\langle \boldsymbol{w}_j^{(t)}, \boldsymbol{\xi}_i \rangle| \right)}_{\in (0,1) \text{ as shown below}}
\end{aligned}
$$

Note that Lemma B.1 gives us:

$$
\begin{aligned}
1 > 1 - \frac{3\rho^{(t)}}{N} l_i^{(t)} \|\boldsymbol{\xi}_i\|^2 |\langle \boldsymbol{w}_j^{(t)}, \boldsymbol{\xi}_i \rangle| &= 1 - \frac{3}{N} \frac{\rho}{\|\nabla \mathcal{L}(\boldsymbol{W}^{(t)})\|_F} l_i^{(t)} \|\boldsymbol{\xi}_i\|^2 |\langle \boldsymbol{w}_j^{(t)}, \boldsymbol{\xi}_i \rangle| \\
&\geq 1 - \frac{3}{N} \frac{\rho}{\frac{3}{N} l_i^{(t)} \langle \boldsymbol{w}_j^{(t)}, \boldsymbol{\xi}_i \rangle^2 \|\boldsymbol{\xi}_i\|} l_i^{(t)} \|\boldsymbol{\xi}_i\|^2 |\langle \boldsymbol{w}_j^{(t)}, \boldsymbol{\xi}_i \rangle| \\
&= 1 - \frac{\rho \|\boldsymbol{\xi}_i\|}{|\langle \boldsymbol{w}_j^{(t)}, \boldsymbol{\xi}_i \rangle|} \\
&> 1 - \frac{|\langle \boldsymbol{w}_j^{(t)}, \boldsymbol{\xi}_i \rangle|}{\|\boldsymbol{\xi}_i\|} \frac{\|\boldsymbol{\xi}_i\|}{|\langle \boldsymbol{w}_j^{(t)}, \boldsymbol{\xi}_i \rangle|} \quad \text{by the choice of } \rho \\
&= 0. \quad (6)
\end{aligned}
$$

Hence, we have that $\operatorname{sgn}(\langle \boldsymbol{w}_{j,\boldsymbol{\epsilon}}^{(t)}, \boldsymbol{\xi}_i \rangle) = \operatorname{sgn}(\langle \boldsymbol{w}_j^{(t)}, \boldsymbol{\xi}_i \rangle)$ and $\boldsymbol{\xi}_i \in \mathcal{I}_{j,\boldsymbol{\epsilon},+}^{(t)}$. Since $\mathcal{I}_{j,+}^{(t)}$ and $\mathcal{I}_{j,-}^{(t)}$ cover all the noises, the lemma statement follows. □

## B.1 PROOF OF THEOREM 4.1.

We then start with item (1) of the theorem. First, for GD, each noise update is computed via:

$$
\begin{aligned}
\langle \boldsymbol{w}_j^{(t+1)}, \boldsymbol{\xi}_i \rangle &= \langle \boldsymbol{w}_j^{(t)}, \boldsymbol{\xi}_i \rangle - \eta \langle \nabla_{\boldsymbol{w}_j^{(t)}} \mathcal{L}(\boldsymbol{W}^{(t)}), \boldsymbol{\xi}_i \rangle \\
&= \langle \boldsymbol{w}_j^{(t)}, \boldsymbol{\xi}_i \rangle + \frac{3\eta}{N} l_i^{(t)} y_i \langle \boldsymbol{w}_j^{(t)}, \boldsymbol{\xi}_i \rangle^2 \|\boldsymbol{\xi}_i\|^2 \quad \text{by Lemma A.6} \\
&= \langle \boldsymbol{w}_j^{(t)}, \boldsymbol{\xi}_i \rangle \left( 1 + \eta \frac{3}{N} l_i^{(t)} y_i \|\boldsymbol{\xi}_i\|^2 \langle \boldsymbol{w}_j^{(t)}, \boldsymbol{\xi}_i \rangle \right) \quad (7)
\end{aligned}
$$

Then by definition, for all the noises $\boldsymbol{\xi}_i \in \mathcal{I}_{j,-}^{(t)}$, $y_i$ and $\langle \boldsymbol{w}_j^{(t)}, \boldsymbol{\xi}_i \rangle$ have opposite signs, so the above equation becomes:

$$\langle \boldsymbol{w}_j^{(t+1)}, \boldsymbol{\xi}_i \rangle = \langle \boldsymbol{w}_j^{(t)}, \boldsymbol{\xi}_i \rangle \left( 1 - \eta \frac{3}{N} l_i^{(t)} \|\boldsymbol{\xi}_i\|^2 |\langle \boldsymbol{w}_j^{(t)}, \boldsymbol{\xi}_i \rangle| \right).$$

Without loss of generality, we consider the case when $\langle \boldsymbol{w}_j^{(t)}, \boldsymbol{\xi}_i \rangle > 0$. We then define the corresponding sequence $a_t = \langle \boldsymbol{w}_j^{(t)}, \boldsymbol{\xi}_i \rangle$ generated by the update:

$$a_{t+1} = a_t (1 - \eta C_i l_i^{(t)} a_t), \quad \text{where } C_i = \frac{3}{N} \|\boldsymbol{\xi}_i\|^2.$$

Next, we want to show that given a proper $\eta$, this sequence is monotonic towards 0, meaning that these noises are gradually "unlearned" by the weights. We provide an inductive step below, which can be easily generalized.

If the learning rate satisfies

$$0 < \eta < \frac{1}{C_i a_t} = \frac{1}{C_i \langle \boldsymbol{w}_j^{(t)}, \boldsymbol{\xi}_i \rangle},$$

then by the update, the following inequality holds:

$$0 < a_t \left( 1 - \frac{1}{C_i a_t} C_i l_i^{(t)} a_t \right) = a_t (1 - l_i^{(t)}) < a_{t+1} < a_t$$

Consequently, $a_{t+1} < a_t$ implies that

$$\eta < \frac{1}{C_i a_t} < \frac{1}{C_i a_{t+1}} \implies a_{t+2} < a_{t+1} \quad \text{by the same argument.}$$

A similar argument holds for $\langle \boldsymbol{w}_j^{(t)}, \boldsymbol{\xi}_i \rangle < 0$. Hence, if we take sufficiently small

$$\eta < \min \left\{ \frac{1}{C_i |\langle \boldsymbol{w}_j^{(t)}, \boldsymbol{\xi}_i \rangle|} : \boldsymbol{\xi}_i \in \mathcal{I}_{j,-}^{(t)} \right\},$$

the weights' alignment with these noises will be monotonic throughout the updates and get closer and closer to 0. The proof for SAM is similar (we replace $\langle \boldsymbol{w}_j^{(t)}, \boldsymbol{\xi}_i \rangle$ by $\langle \boldsymbol{w}_{j,\epsilon}^{(t)}, \boldsymbol{\xi}_i \rangle$ for $\boldsymbol{\xi}_i \in \mathcal{I}_{j,\epsilon,-}^{(t)}$, etc.).

Now we proceed with item (2).

Then we consider $\boldsymbol{\xi}_i \in \mathcal{I}_{j,+}^{(t)} = \mathcal{I}_{j,\epsilon,+}^{(t)}$ in the setting of Lemma B.2. Eq. 7 now becomes:

$$\langle \boldsymbol{w}_j^{(t+1)}, \boldsymbol{\xi}_i \rangle = \langle \boldsymbol{w}_j^{(t)}, \boldsymbol{\xi}_i \rangle \left( 1 + \eta \frac{3}{N} l_i^{(t)} y_i \|\boldsymbol{\xi}_i\|^2 \langle \boldsymbol{w}_j^{(t)}, \boldsymbol{\xi}_i \rangle \right)$$

$$= \langle \boldsymbol{w}_j^{(t)}, \boldsymbol{\xi}_i \rangle \left( 1 + \eta \frac{3}{N} l_i^{(t)} \|\boldsymbol{\xi}_i\|^2 |\langle \boldsymbol{w}_j^{(t)}, \boldsymbol{\xi}_i \rangle| \right) \implies |\langle \boldsymbol{w}_j^{(t+1)}, \boldsymbol{\xi}_i \rangle| > |\langle \boldsymbol{w}_j^{(t)}, \boldsymbol{\xi}_i \rangle|. \quad (8)$$

And a similar expression holds for SAM. Consequently, the alignments will be monotonically away from 0, continually being learned through iterations. Hence, we want to compare how the gradients align with noises: $|\langle \nabla_{\boldsymbol{w}_{j,\epsilon}} \mathcal{L}(\boldsymbol{W}^{(t)} + \boldsymbol{\epsilon}^{(t)}), \boldsymbol{\xi}_i \rangle|$ vs. $|\langle \nabla_{\boldsymbol{w}_j} \mathcal{L}(\boldsymbol{W}^{(t)}), \boldsymbol{\xi}_i \rangle|$. By Lemma A.6, we have that

$$\text{For } \textbf{GD:} \quad |\langle \nabla_{\boldsymbol{w}_j} \mathcal{L}(\boldsymbol{W}^{(t)}), \boldsymbol{\xi}_i \rangle| = \frac{3}{N} l_i^{(t)} \langle \boldsymbol{w}_j^{(t)}, \boldsymbol{\xi}_i \rangle^2 \|\boldsymbol{\xi}_i\|^2.$$

$$\text{For } \textbf{SAM:} \quad |\langle \nabla_{\boldsymbol{w}_{j,\epsilon}} \mathcal{L}(\boldsymbol{W}^{(t)} + \boldsymbol{\epsilon}^{(t)}), \boldsymbol{\xi}_i \rangle| = \frac{3}{N} l_{i,\epsilon}^{(t)} \langle \boldsymbol{w}_{j,\epsilon}^{(t)}, \boldsymbol{\xi}_i \rangle^2 \|\boldsymbol{\xi}_i\|^2$$

$$\overset{(i)}{=} \frac{3}{N} l_{i,\epsilon}^{(t)} \left( \langle \boldsymbol{w}_j^{(t)}, \boldsymbol{\xi}_i \rangle - \frac{3\rho^{(t)}}{N} l_i^{(t)} y_i \langle \boldsymbol{w}_j^{(t)}, \boldsymbol{\xi}_i \rangle^2 \|\boldsymbol{\xi}_i\|^2 \right)^2 \|\boldsymbol{\xi}_i\|^2$$

$$= \frac{3}{N} l_{i,\epsilon}^{(t)} \langle \boldsymbol{w}_j^{(t)}, \boldsymbol{\xi}_i \rangle^2 \left( 1 - \frac{3\rho^{(t)}}{N} l_i^{(t)} y_i \langle \boldsymbol{w}_j^{(t)}, \boldsymbol{\xi}_i \rangle \|\boldsymbol{\xi}_i\|^2 \right)^2 \|\boldsymbol{\xi}_i\|^2$$

$$\overset{(ii)}{=} \frac{3}{N} l_i^{(t)} \langle \boldsymbol{w}_j^{(t)}, \boldsymbol{\xi}_i \rangle^2 \left( 1 - \frac{3\rho^{(t)}}{N} l_i^{(t)} \|\boldsymbol{\xi}_i\|^2 |\langle \boldsymbol{w}_j^{(t)}, \boldsymbol{\xi}_i \rangle| \right)^2 \|\boldsymbol{\xi}_i\|^2,$$

$$(9)$$

where (i) follows from Eq. 4 and (ii) uses the approximation for the logits in early training ($l_i^{(t)} = l_{i,\boldsymbol{\epsilon}}^{(t)}$). Now with a sufficiently small $\rho$ that satisfies Equation 3, the bound in 6 yields

$$|\langle \nabla_{\boldsymbol{w}_{j,\boldsymbol{\epsilon}}} \mathcal{L}(\boldsymbol{W}^{(t)} + \boldsymbol{\epsilon}^{(t)}), \boldsymbol{\xi}_i \rangle| = \frac{3}{N} l_i^{(t)} \langle \boldsymbol{w}_j^{(t)}, \boldsymbol{\xi}_i \rangle^2 \underbrace{\left(1 - \frac{3\rho^{(t)}}{N} l_i^{(t)} \|\boldsymbol{\xi}_i\|^2 |\langle \boldsymbol{w}_j^{(t)}, \boldsymbol{\xi}_i \rangle|\right)^2}_{\in (0,1)} \|\boldsymbol{\xi}_i\|^2$$

$$< \frac{3}{N} l_i^{(t)} \langle \boldsymbol{w}_j^{(t)}, \boldsymbol{\xi}_i \rangle^2 \|\boldsymbol{\xi}_i\|^2 = |\langle \nabla_{\boldsymbol{w}_j} \mathcal{L}(\boldsymbol{W}^{(t)}), \boldsymbol{\xi}_i \rangle|.$$

Since this inequality holds for all $\boldsymbol{\xi}_i \in \mathcal{I}_{j,+}^{(t)} = \mathcal{I}_{j,\boldsymbol{\epsilon},+}^{(t)}$, we compute the average over these noises and have that:

$$\textbf{NoiseAlign}(\mathcal{I}_{j,\boldsymbol{\epsilon},+}^{(t)}, \boldsymbol{w}_{j,\boldsymbol{\epsilon}}^{(t)}) = \textbf{NoiseAlign}(\mathcal{I}_{j,+}^{(t)}, \boldsymbol{w}_{j,\boldsymbol{\epsilon}}^{(t)}) < \textbf{NoiseAlign}(\mathcal{I}_{j,+}^{(t)}, \boldsymbol{w}_j^{(t)}).$$

The special case of the theorem can be proved by directly setting the early training weights to be the initialization $\boldsymbol{W}^{(0)}$. Since $\boldsymbol{W}^{(0)} \sim \mathcal{N}(0, \sigma_0^2)$ and $\boldsymbol{\xi}_i \sim \mathcal{N}(0, \sigma_p^2/d)$, with a large $N$, $\mathcal{I}_{j,+}^{(0)}$ and $\mathcal{I}_{j,-}^{(0)}$ will each contain roughly half of the noises, as $\text{sgn}(\langle \boldsymbol{w}_j^{(0)}, \boldsymbol{\xi}_i \rangle) = \text{sgn}(y_i)$ has probability 0.5 in this case.

**Remark.** The theory matches our insights that the noises tend to be fitted in general for the upsampled dataset. We do note that the bound in Lemma B.1 can be loose, as it covers very extreme cases, and the actual choice of $\rho$ could be much larger in general without breaking the logic of our argument.

## C GENERATION VS. UPSAMPLING: BIAS PERSPECTIVE AND PROOF OF THEOREM 4.2

Item (1) of this theorem can be proved in a similar fashion as for Theorem 4.1 in Appendix B.

For item (2), the update rules follow directly from Lemmas A.2, A.4. Again these formulas themselves can partially explain what happens: the "generation" gradient learns every noise in the same manner, but the "upsampling" gradient learns the replicated noises $k$ times more in one iteration.

**Lemma C.1.** *(Static noise sets) In our setting, with a sufficiently small learning rate $\eta$, for all $j \in [J]$ and all the early iterations $t = 0, \ldots, T$, we have that:*

$$\mathcal{I}_{j,+}^{G,(0)} = \mathcal{I}_{j,+}^{G,(1)} = \cdots = \mathcal{I}_{j,+}^{G,(t)} = \cdots = \mathcal{I}_{j,+}^{G,(T)},$$

$$\mathcal{I}_{j,+}^{U,(0)} = \mathcal{I}_{j,+}^{U,(1)} = \cdots = \mathcal{I}_{j,+}^{U,(t)} = \cdots = \mathcal{I}_{j,+}^{U,(T)}.$$

*The same holds for the corresponding sets $\mathcal{I}_{j,-}^{G,(t)}$ and $\mathcal{I}_{j,-}^{U,(t)}$.*

*Proof.* This directly follows from the previous proofs, where starting from the intializations, small learning rate ensures that certain noise alignments move towards 0 and others move away from 0 early in training. Hence, in this process, the signs do not change, and these sets always contain the same noises as $t$ progresses $\qquad\square$

We now focus on the special case of the theorem and show that in expectation, our requirement on the generation noise prevents **NoiseAlign**$(\mathcal{I}_{j,+}^{G,(t)}, \boldsymbol{w}_j^{(t)})$ from getting too large.

Lemma C.1 implies the sets of noises to which we overfit or do not overfit remain unchanged during early training. Hence, we eliminate the possibility that noises move from one set to another, which simplifies the big picture. The technique is similar to what we did in Theorem 4.1, and we show that this also holds true for data augmented via generation.

For Theorem 4.2, the expectation is taken with respect to the underlying data (and noise) distributions. We first start with computing $\mathbb{E}\left[\textbf{NoiseAlign}(\mathcal{I}_{j,+}^{G,(t)}, \boldsymbol{w}_j^{(t)})\right]$. Expanding using the definition of **NoiseAlign**, we have that this quantity equals:

$$\mathbb{E}\left[\frac{1}{|\mathcal{I}_{j,+}^{G,(t)}|}\left(\sum_{\substack{i=1,\\ \boldsymbol{\xi}_i\in\mathcal{I}_{j,+}^{G,(t)}}}^{N}\left|\langle\nabla_{\boldsymbol{w}_j^{(t)}}\mathcal{L}(\boldsymbol{W}^{(t)}),\boldsymbol{\xi}_i\rangle\right| + \sum_{\substack{i=N+1,\\ \boldsymbol{\gamma}_i\in\mathcal{I}_{j,+}^{G,(t)}}}^{N_{\text{new}}}\left|\langle\nabla_{\boldsymbol{w}_j^{(t)}}\mathcal{L}(\boldsymbol{W}^{(t)}),\boldsymbol{\gamma}_i\rangle\right|\right)\right]$$

$$= \mathbb{E}\left[\left|\langle\nabla_{\boldsymbol{w}_j^{(t)}}\mathcal{L}(\boldsymbol{W}^{(t)}),\boldsymbol{\phi}_i\rangle\right| : \boldsymbol{\phi}_i\in\mathcal{I}_{j,+}^{G,(t)}\right]$$

$$= p\,\mathbb{E}\left[\left|\langle\nabla_{\boldsymbol{w}_j^{(t)}}\mathcal{L}(\boldsymbol{W}^{(t)}),\boldsymbol{\xi}_i\rangle\right| : \boldsymbol{\xi}_i\in\mathcal{I}_{j,+}^{G,(t)}\right] + (1-p)\,\mathbb{E}\left[\left|\langle\nabla_{\boldsymbol{w}_j^{(t)}}\mathcal{L}(\boldsymbol{W}^{(t)}),\boldsymbol{\gamma}_i\rangle\right| : \boldsymbol{\gamma}_i\in\mathcal{I}_{j,+}^{G,(t)}\right]$$

$$= p\,\mathbb{E}\left[\left|\langle\nabla_{\boldsymbol{w}_j^{(t)}}\mathcal{L}(\boldsymbol{W}^{(t)}),\boldsymbol{\xi}\rangle\right|\right] + (1-p)\,\mathbb{E}\left[\left|\langle\nabla_{\boldsymbol{w}_j^{(t)}}\mathcal{L}(\boldsymbol{W}^{(t)}),\boldsymbol{\gamma}\rangle\right|\right], \tag{10}$$

where $p = \frac{N}{N_{new}} = \frac{1}{\alpha+k(1-\alpha)}$ measures the probability that the noise belongs to the original data instead of the synthetic data. The last equality is due to the fact that $\boldsymbol{\phi}_i\in\mathcal{I}_{j,+}^{G,(t)}$ are generated i.i.d and follow the same noise distribution $\mathcal{D}$ or $\mathcal{D}_{\boldsymbol{\gamma}}$.

Recall the following assumption:

$$\mathbb{E}\left[l_{\boldsymbol{\gamma}}(t)\langle\boldsymbol{w}_j^{(t)},\boldsymbol{\gamma}\rangle^2\|\boldsymbol{\gamma}\|^2\right] < (k+1)\mathbb{E}\left[l_{\boldsymbol{\xi}}(t)\langle\boldsymbol{w}_j^{(t)},\boldsymbol{\xi}\rangle^2\|\boldsymbol{\xi}\|^2\right]$$

$$\mathbb{E}\left[\frac{3}{N}l_{\boldsymbol{\gamma}}(t)\langle\boldsymbol{w}_j^{(t)},\boldsymbol{\gamma}\rangle^2\|\boldsymbol{\gamma}\|^2\right] < (k+1)\mathbb{E}\left[\frac{3}{N}l_{\boldsymbol{\xi}}(t)\langle\boldsymbol{w}_j^{(t)},\boldsymbol{\xi}\rangle^2\|\boldsymbol{\xi}\|^2\right]$$

$$\mathbb{E}\left[\left|\langle\nabla_{\boldsymbol{w}_j^{(t)}}\mathcal{L}(\boldsymbol{W}^{(t)}),\boldsymbol{\gamma}\rangle\right|\right] < (k+1)\mathbb{E}\left[\left|\langle\nabla_{\boldsymbol{w}_j^{(t)}}\mathcal{L}(\boldsymbol{W}^{(t)}),\boldsymbol{\xi}\rangle\right|\right], \quad \boldsymbol{\gamma}\sim\mathcal{D}_{\boldsymbol{\gamma}},\ \boldsymbol{\xi}\sim\mathcal{D}$$

where the last inequality follows the expression of noise gradient. Again, note that $\mathcal{I}_{j,+}^{G,(t)}$ contains synthetic noises that follow the distribution $\mathcal{D}_{\boldsymbol{\gamma}}$ since the synthetic data points are generated independently. However, the noises in $\mathcal{I}_{j,+}^{U,(t)}$ are not independent due to upsampling and therefore do not follow the noise distribution in $\mathcal{D}$.

To tackle this issue, we use the gradient alignment bound above and continue with Equation 10

$$< p\,\mathbb{E}\left[\left|\langle\nabla_{\boldsymbol{w}_j^{(t)}}\mathcal{L}(\boldsymbol{W}^{(t)}),\boldsymbol{\xi}\rangle\right|\right] + (1-p)(k-1)\,\mathbb{E}\left[\left|\langle\nabla_{\boldsymbol{w}_j^{(t)}}\mathcal{L}(\boldsymbol{W}^{(t)}),\boldsymbol{\xi}\rangle\right|\right]$$

$$= \frac{1}{\alpha+k(1-\alpha)}\,\mathbb{E}\left[\left|\langle\nabla_{\boldsymbol{w}_j^{(t)}}\mathcal{L}(\boldsymbol{W}^{(t)}),\boldsymbol{\xi}\rangle\right|\right] + \frac{(k^2-1)(1-\alpha)}{\alpha+k(1-\alpha)}\,\mathbb{E}\left[\left|\langle\nabla_{\boldsymbol{w}_j^{(t)}}\mathcal{L}(\boldsymbol{W}^{(t)}),\boldsymbol{\xi}\rangle\right|\right]$$

$$= \frac{\alpha+(1-\alpha)}{\alpha+k(1-\alpha)}\,\mathbb{E}\left[\left|\langle\nabla_{\boldsymbol{w}_j^{(t)}}\mathcal{L}(\boldsymbol{W}^{(t)}),\boldsymbol{\xi}\rangle\right|\right] + \frac{(k^2-1)(1-\alpha)}{\alpha+k(1-\alpha)}\,\mathbb{E}\left[\left|\langle\nabla_{\boldsymbol{w}_j^{(t)}}\mathcal{L}(\boldsymbol{W}^{(t)}),\boldsymbol{\xi}\rangle\right|\right]$$

$$= \frac{\alpha}{\alpha+k(1-\alpha)}\,\mathbb{E}\left[\left|\langle\nabla_{\boldsymbol{w}_j^{(t)}}\mathcal{L}(\boldsymbol{W}^{(t)}),\boldsymbol{\xi}\rangle\right|\right] + \frac{k^2(1-\alpha)}{\alpha+k(1-\alpha)}\,\mathbb{E}\left[\left|\langle\nabla_{\boldsymbol{w}_j^{(t)}}\mathcal{L}(\boldsymbol{W}^{(t)}),\boldsymbol{\xi}\rangle\right|\right]$$

$$= \mathbb{E}\left[\frac{1}{|\mathcal{I}_{j,+}^{U,(t)}|}\left(\sum_{\substack{i=1,\\ \boldsymbol{\xi}_i\in\mathcal{I}_{j,+}^{U,(t)}}}^{\alpha N}\left|\langle\nabla_{\boldsymbol{w}_j^{(t)}}\mathcal{L}(\boldsymbol{W}^{(t)}),\boldsymbol{\xi}_i\rangle\right| + k\sum_{\substack{i=\alpha N+1,\\ \boldsymbol{\xi}_i\in\mathcal{I}_{j,+}^{U,(t)}}}^{N}k\left|\langle\nabla_{\boldsymbol{w}_j^{(t)}}\mathcal{L}(\boldsymbol{W}^{(t)}),\boldsymbol{\xi}_i\rangle\right|\right)\right]$$

$$= \mathbb{E}\left[\textbf{NoiseAlign}(\mathcal{I}_{j,+}^{U,(t)},\boldsymbol{w}_j^{(t)})\right].$$

The last equality follows from the distribution of noises in the upsampled dataset: the slow-learnable $(1-\alpha)N$ data points are copied $k$ times, and from the gradient computation (first part of this theorem), each is learned $k$ times more, giving rise to the two $k$'s. The noises in each summation term follow $\mathcal{D}$, as they are independent within the subset of summands.

Hence, we have shown the desired result:

$$\mathbb{E}\left[\textbf{NoiseAlign}(\mathcal{I}_{j,+}^{G,(t)},\boldsymbol{w}_j^{(t)})\right] < \mathbb{E}\left[\textbf{NoiseAlign}(\mathcal{I}_{j,+}^{U,(t)},\boldsymbol{w}_j^{(t)})\right].$$

At a high level, this suggests that starting at the weight $\boldsymbol{w}_j^{(t)}$, as long as the notion of small synthetic noise is satisfied, the expected gradient alignment with noises in the set where overfitting occurs is strictly lower when the dataset is augmented through generation. This reduced expected **NoiseAlign** leads to gradient updates that place less emphasis on fitting noise, thereby enabling the model to focus more effectively on learning the true features.

Consequently, although both upsampling and generation promote more uniform learning, their distinct interactions with noise critically influence generalization performance. This highlights the necessity of incorporating synthetic data and motivates our method.

# D  GENERATION VS. UPSAMPLING: VARIANCE PERSPECTIVE AND PROOF OF THEOREM 4.3

For generation, we have a fully independent dataset, so given the per-gradient variance bound $\sigma_G^2$, the variance of the mini-batch gradient $\hat{\boldsymbol{g}}_G$ can be computed as:

$$
\begin{aligned}
\mathbb{E}\left[\|\hat{\boldsymbol{g}}_G - \bar{\boldsymbol{g}}\|^2\right] = \mathbb{E}\left[\|\frac{1}{B}\sum_{i=1}^{B}\boldsymbol{g}_i - \bar{\boldsymbol{g}}\|^2\right] &= \mathbb{E}\left[\|\frac{1}{B}\sum_{i=1}^{B}\boldsymbol{g}_i - \frac{1}{B}B\bar{\boldsymbol{g}}\|^2\right] \\
&= \frac{1}{B^2}\mathbb{E}\left[\|\sum_{i=1}^{B}(\boldsymbol{g}_i - \bar{\boldsymbol{g}})\|^2\right] \\
&= \frac{1}{B^2}\sum_{i=1}^{B}\mathbb{E}\left[\|\boldsymbol{g}_i - \bar{\boldsymbol{g}}\|^2\right] \\
&\leq \frac{1}{B^2}B\sigma^2 \\
&= \frac{\sigma^2}{B},
\end{aligned}
\tag{11}
$$

where we can directly take the summation out of expectation due to independence among the data.

For the upsampled dataset, we instead have two parts:

1. **Fully independent part:** The first $\alpha N$ data points that contain the fast-learnable feature are independent from each other and the rest.

2. **Replicated part:** The rest of the $k(1-\alpha)N$ data points that contain duplicates. This part introduces the risk that the same data points might be selected into the same batch.

With this motivation, we let $B = B_1 + B_2$, where $B_1$ denotes the number of data points selected from the fully independent part, and $B_2$ denotes the number selected from the replicated part.

For theoretical simplicity, we consider a stratified mini-batch gradient, where $B_1$ and $B_2$ are fixed numbers in proportion to the sizes of the two parts. We assume that $B_1$, $B_2$ are both integers such that they follow the proportion:

$$
B_1 = B\frac{\alpha N}{\alpha N + k(1-\alpha)N} = B\frac{\alpha}{\alpha + k(1-\alpha)}.
\tag{12}
$$

$$
B_2 = B\frac{k(1-\alpha)N}{\alpha N + k(1-\alpha)N} = B\frac{k(1-\alpha)}{\alpha + k(1-\alpha)}.
\tag{13}
$$

Note that this removes one source of variance since $B_1$ and $B_2$ should themselves be random variables in the actual gradient. Hence, the actual variance should be larger than that of the stratified version.

The variance of the stratefied mini-batch gradient $\hat{\boldsymbol{g}}_U$ after upsampling can be split as:

$$
\mathbb{E}\left[\|\hat{\boldsymbol{g}}_U - \bar{\boldsymbol{g}}\|^2\right] = \mathbb{E}\left[\|\frac{1}{B}\sum_{i\in B}\boldsymbol{g}_i - \frac{1}{B}B\bar{\boldsymbol{g}}\|^2\right] = \frac{1}{B^2}\mathbb{E}\left[\|\sum_{i\in B_1}(\boldsymbol{g}_i - \bar{\boldsymbol{g}})\|^2\right] + \frac{1}{B^2}\mathbb{E}\left[\|\sum_{i\in B_2}(\boldsymbol{g}_i - \bar{\boldsymbol{g}})\|^2\right]
\tag{14}
$$

due to independence and unbiasedness. Here with a slight abuse of notation, $i \in B, B_1, B_2$ represents the indices of data in each portion. Similar to Equation 11, we have for the independent part,

$$\frac{1}{B^2}\mathbb{E}\left[\|\sum_{i \in B_1}(g_i - \bar{g})\|^2\right] \leq \frac{\sigma^2}{B^2}B_1 = \frac{\sigma^2}{B}\frac{\alpha}{\alpha + k(1 - \alpha)}. \tag{15}$$

For the replicated part, since we may select copied data points, we first rewrite the summation as:

$$\sum_{i \in B_2}g_i = \sum_{i=1}^{(1-\alpha)N}Y_i g_i,$$

where we index over all the unique data points $i \in B_2$ and introduce the random variable $Y_i \geq 0$ representing the number of copies of each unique point, subject to $\sum_{i=1}^{(1-\alpha)N}Y_i = B_2$.

Clearly, as we are selecting $B_2$ data points from a total of $k(1 - \alpha)N$ points $((1 - \alpha)N$ unique ones each with $k$ copies), $Y_i$ follows a hypergeometric distriution. From classical probability theory, we have:

1. Mean:

$$\mathbb{E}[Y_i] = B_2\frac{k}{k(1 - \alpha)N} = B\frac{k(1 - \alpha)}{\alpha + k(1 - \alpha)}\frac{k}{k(1 - \alpha)N} = B\frac{k}{\alpha N + k(1 - \alpha)N}.$$

2. Variance:

$$\begin{aligned}
\mathrm{Var}(Y_i) &= \mathbb{E}[Y_i]\frac{k(1 - \alpha)N - k}{k(1 - \alpha)N}\frac{k(1 - \alpha)N - B_2}{k(1 - \alpha)N - 1} \\
&= \mathbb{E}[Y_i]\frac{(1 - \alpha)N - 1}{(1 - \alpha)N}\frac{k(1 - \alpha)N - B\frac{k(1-\alpha)}{\alpha+k(1-\alpha)}}{k(1 - \alpha)N - 1} \\
&= \mathbb{E}[Y_i]\frac{(1 - \alpha)N - 1}{(1 - \alpha)N}\frac{k(1 - \alpha)(\alpha N + k(1 - \alpha)N - B)}{(k(1 - \alpha)N - 1)(\alpha + k(1 - \alpha))} \\
&= \mathbb{E}[Y_i]\frac{(1 - \alpha)N - 1}{N}\frac{k(\alpha N + k(1 - \alpha)N - B)}{(k(1 - \alpha)N - 1)(\alpha + k(1 - \alpha))} \\
&= \mathbb{E}[Y_i]\frac{k(1 - \alpha)N - k}{k(1 - \alpha)N - 1}\frac{\alpha N + k(1 - \alpha)N - B}{\alpha N + k(1 - \alpha)N} \\
&\leq \mathbb{E}[Y_i]\frac{\alpha N + k(1 - \alpha)N - B}{\alpha N + k(1 - \alpha)N} \\
&= B\frac{k}{\alpha N + k(1 - \alpha)N}\left(1 - \frac{B}{\alpha N + k(1 - \alpha)N}\right).
\end{aligned}$$

3. Second Moment:

$$\begin{aligned}
\mathbb{E}[Y_i^2] &= \mathrm{Var}(Y_i) + \mathbb{E}[Y_i]^2 \\
&\leq B\frac{k}{\alpha N + k(1 - \alpha)N}\left(1 - \frac{B}{\alpha N + k(1 - \alpha)N}\right) + \left(B\frac{k}{\alpha N + k(1 - \alpha)N}\right)^2 \\
&= B\frac{k}{\alpha N + k(1 - \alpha)N}\left(1 - \frac{B}{\alpha N + k(1 - \alpha)N} + \frac{Bk}{\alpha N + k(1 - \alpha)N}\right) \\
&= B\frac{k}{\alpha N + k(1 - \alpha)N}\left(1 + \frac{(k - 1)B}{\alpha N + k(1 - \alpha)N}\right). \tag{16}
\end{aligned}$$

With these quantities, we can now compute:

$$\frac{1}{B^2}\mathbb{E}\left[\|\sum_{i \in B_2}(\boldsymbol{g}_i - \bar{\boldsymbol{g}})\|^2\right] = \frac{1}{B^2}\mathbb{E}\left[\|\sum_{i=1}^{(1-\alpha)N} Y_i(\boldsymbol{g}_i - \bar{\boldsymbol{g}})\|^2\right] \quad \text{since} \sum_{i=1}^{(1-\alpha)N} Y_i = B_2$$

$$= \frac{1}{B^2}\sum_{i=1}^{(1-\alpha)N}\mathbb{E}\left[Y_i^2\|\boldsymbol{g}_i - \bar{\boldsymbol{g}}\|^2\right]$$

$$\leq \frac{\sigma^2}{B^2}\sum_{i=1}^{(1-\alpha)N}\mathbb{E}\left[Y_i^2\right]$$

$$= \frac{\sigma^2}{B}\frac{k(1-\alpha)N}{\alpha N + k(1-\alpha)N}\left(1 + \frac{(k-1)B}{\alpha N + k(1-\alpha)N}\right) \quad \text{by Equation 16.}$$

$$(17)$$

We plug in Equations 15, 17 into Equation 14 to obtain:

$$\mathbb{E}\left[\|\hat{\boldsymbol{g}}_U - \bar{\boldsymbol{g}}\|^2\right] \leq \frac{\sigma^2}{B}\frac{\alpha}{\alpha + k(1-\alpha)} + \frac{\sigma^2}{B}\frac{k(1-\alpha)}{\alpha + k(1-\alpha)}\left(1 + \frac{(k-1)B}{\alpha N + k(1-\alpha)N}\right)$$

$$= \frac{\sigma^2}{B}\left(1 + \frac{k(k-1)(1-\alpha)}{(\alpha + k(1-\alpha))^2}\frac{B}{N}\right) = I.$$

.

In the theorem statement, we use $\sigma_U^2(k)$ and $\sigma_G^2(k)$ to differentiate the two and emphasize the dependence on the augmenting factor $k$.

# E  PSEUDOCODE

Algorithm 1 illustrates our method. We include Table 3 to compare it with other augmentation methods, including classical ones, naive diffusion-based generation, and mix-based methods such as mixup (Zhang et al. (2017)) and CutMix (Yun et al. (2019)).

# F  ADDITIONAL EXPERIMENTS

## F.1  ADDITIONAL EXPERIMENTAL SETTINGS

**Datasets.** The CIFAR10 dataset consists of 60,000 $32 \times 32$ color images in 10 classes, with 6000 images per class. The CIFAR100 dataset is just like the CIFAR10, except it has 100 classes containing 600 images each. For both of these datasets, the training set has 50,000 images (5,000 per class for CIFAR10 and 500 per class for CIFAR100) with the test set having 10,000 images. Tiny-ImageNet comprises 100,000 images distributed across 200 classes of ImageNet (Deng et al., 2009), with each class containing 500 images. These images have been resized to 64×64 dimensions and are in color. The dataset consists of 500 training images, 50 validation images, and 50 test images per class. For transfer learning experiments, we also used three fine-grained classification datasets. Flowers-102 (Nilsback & Zisserman, 2008) contains 8,189 images of flowers from 102 categories, with between 40 and 258 images per class. Aircraft (Maji et al., 2013) consists of 10,000 images across 100 aircraft model variants, with roughly 100 images per class. Stanford Cars (Krause et al., 2013) contains 16,185 images of 196 classes of cars, with classes defined at the level of make, model, and year.

**Training on different datasets.** From the setting in (Andriushchenko & Flammarion, 2022), we trained Pre-Activation ResNet18 on all datasets for 200 epochs with a batch size of 128. We used SGD with the momentum parameter 0.9 and set weight decay to 0.0005. We also fixed $\rho = 0.1$ for SAM in all experiments unless explicitly stated. We used a linear learning rate schedule starting at 0.1. The learning rate is decayed by a factor of 10 after 50% and 75% epochs, i.e., we set the learning rate to 0.01 after 100 epochs and to 0.001 after 150 epochs. For transfer learning experiments, we fine-tuned a pre-trained ResNet18 on ImageNet-1K. On Flowers-102, we trained for 200 epochs with

Table 3: Comparison with other augmentation methods.

| Feature | Classical Aug (Flip, Crop, etc.) | Mix-based (mixup/CutMix) |
|---|---|---|
| Core Objective | Learn basic invariances | Model regularization |
| Selection Strategy | Applied randomly to all images | Random pairing of images; augments all samples |
| Mechanism | Geometric/color transformations | Mix patches between real images |
| Computation | Negligible: native GPU operations | Low: simple arithmetic operations |
| Guarantee (Theory) | No | Yes |
| Primary Application | Image classification; transfer learning | Image classification; transfer learning |
| Key Advantage | Fast, effective baseline | Cheap and effective regularizer |

| Feature | Diffusion-based | TADA |
|---|---|---|
| Core Objective | Increase data quantity and diversity | Reduce simplicity bias; promote balanced feature learning |
| Selection Strategy | — | Selects samples not learned early in training |
| Mechanism | Generates synthetic images and mixes them with real images | Generates synthetic images close to real images |
| Computation | High: generates multiple times for different conditional prompts | Moderate: only generates a subset (30–40% of dataset) |
| Guarantee (Theory) | No | Yes |
| Primary Application | Image classification; transfer learning | Image classification; transfer learning |
| Key Advantage | Can boost performance at a large compute cost | Efficient: high gain from few samples; can be combined with prior methods |

an initial learning rate of 0.001. On Aircraft and Stanford Cars, we trained for 100 epochs with an initial learning rate of 0.01.

**Training with different architectures.** We used the same training procedures for Pre-Activation ResNet18, VGG19, and DenseNet121. We directly used the official Pytorch (Paszke et al., 2019) implementation for VGG19 and DenseNet121. For ViT (Yuan et al., 2021), we adopted a Pytorch implementation at `https://github.com/lucidrains/vit-pytorch`. In particular, the hidden size, the depth, the number of attention heads, and the MLP size are set to 768, 8, 8, and 2304, respectively. We adjusted the patch size to 4 to fit the resolution of CIFAR10 and set both the initial learning rate and $\rho$ to 0.01. For both ConvNeXt-T and Swin-T, we used their official implementations but trained with SGD instead of Adam. The (learning rate, $\rho$) is set to (0.1, 0.05) for ConvNeXt-T and (0.01, 0.001) for Swin-T.

**Hyperparameters.** For USEFUL (Nguyen et al., 2024), we adopt their default hyper-parameters for separating epochs and set the upsampling factor $k$ to 2 as it yields the best performance. For our method, the upsampling factor $k$ is set to 5 for CIFAR10 and CIFAR100 while that of TinyImageNet is set to 4. Table 4 summarizes the upsampling factor along with the denoising steps at which the synthetic images are used to augment the base training set. For transfer learning experiments, we set $k$ to 2.

**Generating synthetic data with diffusion model.** We use an open-source text-to-image diffusion model, GLIDE (Nichol et al., 2021) as our baseline for image generation. We directly used the official Pytorch implementation at `https://github.com/openai/glide-text2im`. While generating, we set the time steps as 100 and the guidance scale as 3.0. For a k-way classification, we input the class names $C = \{c_1, \ldots, c_k\}$ with prompt $l =' $ a photo of $\{c_i\}'$. For DiffuseMix, we used their official implementation with a single conditional prompt ("Autumn").

---

**Algorithm 1** TADA (**TA**rgeted **D**iffusion **A**ugmentation)

---

1: **Input:** Original dataset $D$, Model $f(\cdot, W^{(0)})$, Separation epoch $t$, Total epochs $T$, Time steps $t_\star$, GLIDE model $G(\mu_\theta, \Sigma_\theta)$
2: **Step 1: Initial Training**
3: Train the model $f(\cdot, W^{(0)})$ on dataset $D$ for $t$ epochs
4: **Step 2: Clustering and Data Selection Based on Loss**
5: **for** each class $c \in D$ **do**
6:     $\{C_1, C_2\} \leftarrow$ k-means$(f(x_j; W^{(t)}))$ {Cluster data into $C_1$ and $C_2$}
7:     **Step 3: Image Generation from Selected Data**
8:     Obtain text prompt $l$ {e.g. For class *dog*, $l$ is *a photo of dog*}
9:     **for** each data point $x^{\text{ref}} \in C_2$ **do**
10:        Initialize random noise $\epsilon \sim \mathcal{N}(0, I)$
11:        Generate initial noisy image $x_{t_\star} \leftarrow \sqrt{\bar{\alpha}_{t_\star}} x^{\text{ref}} + \sqrt{1 - \bar{\alpha}_{t_\star}} \epsilon$
12:        **for** $s$ from $t_\star$ to 1 **do**
13:          $\mu, \Sigma \leftarrow \mu_\theta(x_s, s, l), \Sigma_\theta(x_s, s, l)$
14:          $x_{s-1} \leftarrow$ sample from $\mathcal{N}(\mu, \Sigma)$
15:        **end for**
16:        $x_{\text{generated}} = x_0$
17:        $D = D \cup x_{\text{generated}}$ {Add the generated image to the dataset}
18:     **end for**
19: **end for**
20: **Step 4: Retraining the Model**
21: Train the model $f(\cdot, W)$ on updated dataset $D$ for $T$ epochs
22: **Output:** Final model $f(\cdot, W(T))$

---

Table 4: Details of the upsampling factor $k$ for different methods and datasets.

| Method | Dataset | k | Denoising steps |
|--------|---------|---|-----------------|
| USEFUL | CIFAR10 | 2 | |
| | CIFAR100 | 2 | |
| | Tiny-ImageNet | 2 | |
| TADA | CIFAR10 | 5 | 40, 50, 70, 80 |
| | CIFAR100 | 5 | 40, 50, 70, 80 |
| | Tiny-ImageNet | 4 | 50, 70, 80 |

**Computational resources.** We used 1 NVIDIA RTX 3090 GPU for model training and 4 NVIDIA RTX A5000 GPUs for generating.

## F.2 ADDITIONAL EXPERIMENTAL RESULTS

**State-of-the-art architectures.** To further demonstrate the effectiveness of our method, we evaluated it on two widely used modern vision backbones: ConvNeXt-T (Liu et al., 2022), a convolutional network that revisits ConvNet design with architectural refinements inspired by transformers, and Swin-T (Liu et al., 2021), a hierarchical vision transformer that introduces shifted windows for efficient and scalable self-attention. Both models were trained on CIFAR-10 using SGD with $k = 2$. As shown in Table 1, our method consistently outperforms both the Original (no augmentation) and USEFUL. In particular, for ConvNeXt-T, our approach achieves a nearly 7% reduction in test error compared to the second-best method, highlighting its strong compatibility with modern architectures.

**Qualitative results.** Figure 7 presents examples of slow-learnable (top) and fast-learnable (bottom) samples in CIFAR-10 where the slow-learnable samples are specifically targeted in our synthetic data augmentation. Slow-learnable samples are often visually ambiguous: the object may be partially visible, relatively small compared to the background, or blended with clutter, making them harder to recognize. In contrast, fast-learnable samples are clear and representative of their class, with the object occupying most of the image and little background interference. These characteristics explain

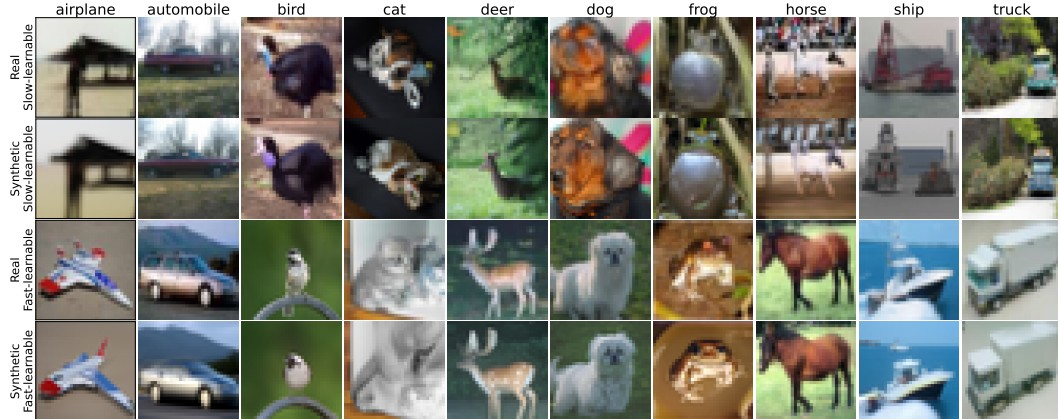

Figure 7: Examples of slow-learnable (row 1) and fast-learnable (row 3) samples in CIFAR-10. Rows 2 and 4 show the corresponding synthetic images generated with GLIDE (Section 4.4) using 50 denoising steps (FID = 10.67).

Table 5: **Targeted augmentation with non-diffusion and diffusion methods.** Test classification error (%) of training ResNet18 on CIFAR10. Here, TrivialAugment (TA) is applied only to the upsampled subset (either slow-learnable or full data) before being added back to the original dataset, which differs from Figure 5 where TA is applied online to all training data. Targeted TA augmentation outperforms full-data TA augmentation, and combining targeted TA augmentaiton with diffusion-generated images yields the best performance.

| Method | Test Error (%) |
|---|---|
| Original | $5.07 \pm 0.04$ |
| TA on slow-learnable examples only | $\mathbf{4.26 \pm 0.02}$ |
| TA on all examples | $4.48 \pm 0.08$ |
| TA + Diffusion on slow-learnable examples only | $\mathbf{3.81 \pm 0.10}$ |
| TA + Diffusion on all examples | $4.20 \pm 0.13$ |

why slow-learnable samples are acquired later in training, whereas fast-learnable ones are learned quickly. Figure 7 shows that our synthetic examples preserve key visual content (features)—such as object shape, structure, and spatial arrangement while introducing subtle variations in texture or color (noise). For instance, an image of a bird in the second row is reproduced with the same layout and pose, but with different color for the neck part. Such nuanced transformations are difficult to achieve with standard augmentations like random cropping or flipping, highlighting the value of generative augmentation in our approach.

**Application to non-diffusion augmentation methods.** Our findings on which examples should be augmented generalize beyond diffusion-based augmentation. To verify this, we conducted additional experiments using TrivialAugment (TA) (Müller & Hutter, 2021), a widely used non-diffusion augmentation method. Specifically, we either augment with only the slow-learnable examples or with the full dataset, apply TA to the augmented copies, and then add them back to the original training set. As shown in Table5, augmenting only slow-learnable examples with TA outperforms applying TA to all examples. Furthermore, combining TA with targeted diffusion-based augmentation yields additional improvements and surpasses full-data augmentation with both TA and synthetic data.

We note that this setting differs from the experiments in Figure 5. In Figure 5, TA is applied online to both original and augmented data during training. In contrast, in Table 5, TA is applied only to the augmented subset (either slow-learnable or full data) before adding it to the original dataset, allowing us to isolate the benefit of targeted example selection.

**Effect of the upsampling factors.** Table 6 illustrates the performance of our method and USE-FUL when varying the upsampling factors. As discussed in Section 4, USEFUL achieves the best

Table 6: Training ResNet18 on CIFAR10 with different upsampling factor (k).

| K | CIFAR10 (USEFUL) | CIFAR10 (TADA) | CIFAR100 (TADA) | TINY-IMAGENET (TADA) |
|---|---|---|---|---|
| 2 | **4.79** | 4.57 | 21.13 | 32.90 |
| 3 | 5.01 | 4.52 | 21.27 | 30.88 |
| 4 | 5.04 | 4.45 | 20.26 | **30.64** |
| 5 | 4.98 | **4.42** | **20.00** | 31.28 |

Table 7: FID and test classification error of ResNet18 on CIFAR10 when augmenting with synthetic images from different denoising steps. The upsampling factor $k$ is set to 2 here.

| STEPS | FID | ERM | SAM |
|---|---|---|---|
| 0(ORIGINAL IMAGES) | | 5.07 | 4.01 |
| 10 | 5.36 | 4.94 | 4.03 |
| 20 | 6.86 | 4.85 | 3.92 |
| 30 | 7.48 | 4.83 | 3.79 |
| 40 | 8.52 | 4.73 | 3.92 |
| 50 | 10.67 | **4.56** | **3.69** |
| 60 | 13.25 | 4.68 | 3.72 |
| 70 | 17.46 | 4.91 | 3.84 |
| 80 | 24.27 | 4.83 | 3.98 |
| 90 | 34.69 | 4.87 | 3.86 |
| 100 | 47.35 | 4.99 | 4.28 |

Table 8: Training ResNet18 on CIFAR10 with different data selection strategy. We used SGD and set the upsampling factor $k$ to 5 here.

| METHOD | MISCLASSIFICATION | HIGH LOSS | SLOW-LEARNABLE (USEFUL AND TADA) |
|---|---|---|---|
| TEST ERROR | $4.87 \pm 0.10$ | $4.77 \pm 0.06$ | **$4.42 \pm 0.04$** |

performance at $k = 2$ due to overfitting noise at larger $k$. On the other hand, using synthetic images, our method benefits from larger values of $k$, yielding the best performance at $k = 5$ for the CIFAR datasets and $k = 4$ for the Tiny-ImageNet dataset. This empirical result reinforces our theoretical findings in Section 4.2.

**Effect of the number of denoising steps.** Table 7 illustrates the impact of the number of denoising steps on both the FID score and the downstream performance of our method. When initializing the denoising process with real images, increasing the number of steps leads to a higher FID score. Unlike prior works in synthetic data augmentation, we observe that minimizing the FID score between real and synthetic images does not necessarily correlate with improved performance. Specifically, using fewer steps generates images that are overly similar to real images, potentially amplifying the inherent noise present in the original data. Conversely, too many denoising steps introduce excessive noise, which also degrades performance. Empirically, we find that using 50 denoising steps strikes the best balance, yielding optimal results for both SGD and SAM optimizers.

**Alternative selection strategies.** Our approach differs from core-set selection methods Guo et al. (2022), which aim to reduce dataset size while matching the performance of training on the full data. Such methods either require training the model (or a smaller proxy) and use its statistics to find the coreset or update the coreset during the training. In contrast to coreset selection, the goal of our approach is to improve the generalization performance over that of full data, by reducing the simplicity bias of training. We showed that this can be done by augmenting the data with faithful synthetic examples corresponding to the slow-learnable part of the data. Motivated by theory, we found such examples by clustering the model output early in training (without training a proxy model or updating the subset during training). However, other more heuristic approaches can be also used to find slow-learnable examples. In this section, we conducted new experiments to find slow-learnable examples based on (1) high loss and (2) misclassification, at the same checkpoint as used in our

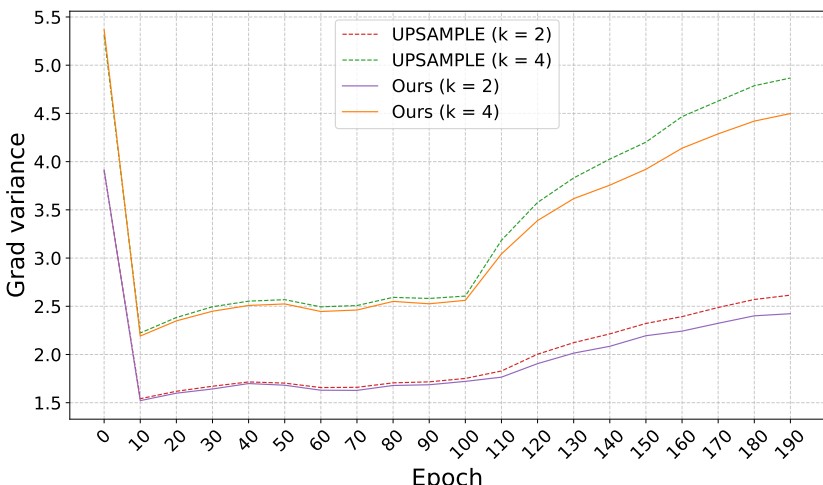

Figure 8: Variance of mini-batch gradients of ResNet18 with SGD on the augmented CIFAR10 dataset found by USEFUL and TADA when varying the upsampling factor $k$. Our method yields lower gradient variance throughout the entire training process.

experiments in the paper. As shown Table 8, our method outperforms both heuristic-based selection strategies. Corset selection methods can be applied on top of our synthetically augmented data to further improve data-efficiency. This is an exciting direction we leave for future work.

**Generation time.** Using 4 GPUs, the generation time for the entire CIFAR-10, CIFAR-100, and Tiny ImageNet datasets is approximately 12, 12, and 26 hours, respectively. In contrast, our method requires generating only about 30%, 40%, and 60% of the total dataset size, reducing the generation time to just 3.6, 4.8, and 15.6 hours, respectively. This represents a substantial improvement in efficiency compared to prior approaches to synthetic data generation, which typically require producing 10–30 times more samples than the original dataset size.

**Variance of mini-batch gradients.** Figure 8 presents the mini-batch gradient variance during training of ResNet18 using SGD on the augmented CIFAR-10 datasets selected by USEFUL and our proposed method. Across all upsampling factors $k$, our method consistently yields lower gradient variance throughout the entire training process. This indicates that our synthetic data provides a more stable and consistent learning signal, which can contribute to more effective optimization. Notably, the variance gap between our method and USEFUL widens significantly after epochs 100 and 150—key points at which the learning rate is decayed. This suggests that our method enables the model to adapt more smoothly to changes in the learning rate, likely due to the benign noise introduced in the faithful synthetic images. Lower gradient variance is often associated with more stable convergence and improved generalization, highlighting the advantage of our approach.

**Training loss.** Figure 9 illustrates that training with synthetic augmentation (TADA) has lower training loss than training with real augmentation (UPSAMPLE).

**Different number of clusters.** Figure 10 illustrates the performance of our method with varying numbers of clusters (2, 3, and 4). For each number of clusters, we varied the number of augmented clusters, which are selected as those with the highest average losses. The baseline (0%) corresponds to no augmentation, while 100% indicates augmenting all clusters. As shown in the figure, augmenting too few or too many examples is suboptimal. For ResNet18 on CIFAR-10, augmenting approximately 30% of the dataset yields the best performance.

### F.3   EXPERIMENTS ON IMAGENET

To demonstrate the scalability of our method, we apply it to training on ImageNet (Deng et al., 2009), a large-scale dataset containing over 1.2M images across 1,000 classes and commonly

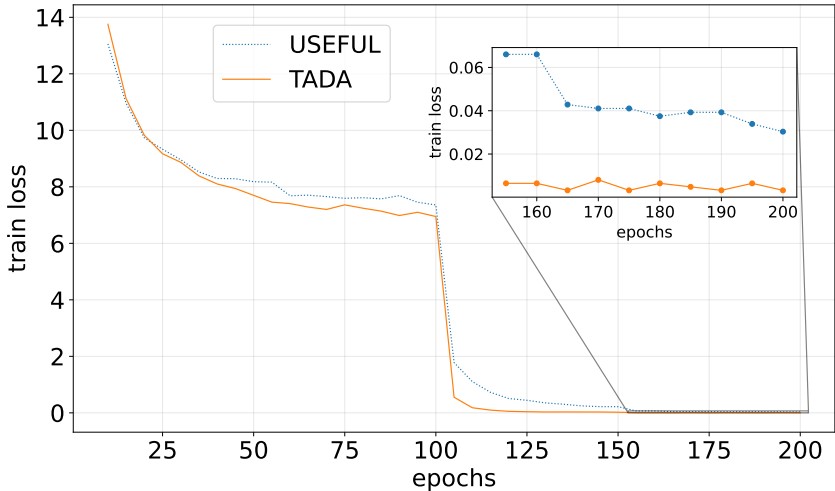

Figure 9: Training loss of ResNet18 on the CIFAR10 dataset when upsampling the slow-learnable examples vs. augmenting them with synthetic data

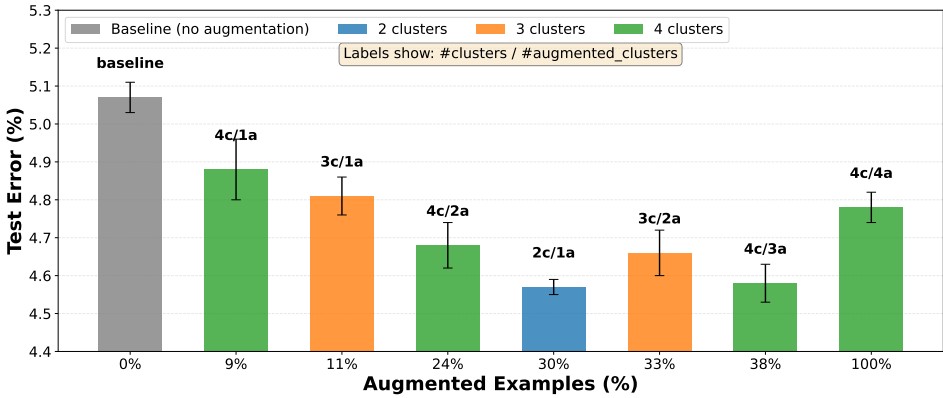

Figure 10: Test error of ResNet18 trained on CIFAR-10 with different cluster-based augmentation configurations. Each bar represents one experimental setting, color-coded by the number of clusters: blue for 2 clusters, orange for 3 clusters, and green for 4 clusters. Augmented clusters are selected as those with the highest average losses. The x-axis shows the percentage of augmented examples relative to the full dataset size. Error bars represent standard deviations across runs.

used as a benchmark for evaluating visual recognition models. Due to computational constraints, we adopt the FFCV library (Leclerc et al., 2022) for efficient data loading and training, adapting the implementation from the publicly available repository at `https://github.com/libffcv/ffcv-imagenet`. We train ResNet18 and ResNet50 using the smallest configuration provided in that codebase (i.e., 16 epochs). For synthetic image generation, we employ Boomerang (Luzi et al., 2022) with Stable Diffusion v1.5 (`https://huggingface.co/stable-diffusion-v1-5/stable-diffusion-v1-5`), following the settings described in Boomerang's paper. For selecting augmented images, we use the checkpoint at epoch 2 and the number of augmented images is about 65% of the full dataset.

Table 9 reports the ImageNet test accuracy when training ResNet18 and ResNet50 with different augmentation strategies. Our method achieves the highest Top-1 and Top-5 accuracies across both architectures, outperforming Boomerang despite using only 65% augmentation, whereas Boomerang

Table 9: Test accuracy of training ResNet18 and ResNet50 on ImageNet.

| Method | Augmentation (%) | ResNet18 | | ResNet50 | |
|---|---|---|---|---|---|
| | | Top-1 Acc | Top-5 Acc | Top-1 Acc | Top-5 Acc |
| Original | 0 | $67.23 \pm 0.02$ | $87.76 \pm 0.10$ | $73.24 \pm 0.02$ | $91.63 \pm 0.08$ |
| USEFUL | 65 | $68.86 \pm 0.20$ | $88.68 \pm 0.12$ | $74.04 \pm 0.15$ | $92.06 \pm 0.08$ |
| Boomerang | 100 | $68.95 \pm 0.04$ | $89.02 \pm 0.09$ | $73.72 \pm 0.06$ | $92.09 \pm 0.11$ |
| TADA+ Boomerang | 65 | $\mathbf{69.28 \pm 0.06}$ | $\mathbf{89.21 \pm 0.09}$ | $\mathbf{74.77 \pm 0.05}$ | $\mathbf{92.62 \pm 0.07}$ |

Table 10: Performance comparison of training YOLOv5m on MS-COCO.

| Method | Augmentation (%) | $AP_{50}$ | $mAP_{50-95}$ |
|---|---|---|---|
| Original | 0 | 63.26 | 44.26 |
| USEFUL | 75 | 63.87 | 44.92 |
| InstanceAugmentation | 100 | 64.17 | 45.75 |
| TADA+ InstanceAugmentation | 75 | **64.94** | **46.28** |

requires 100% synthetic augmentation. This demonstrates that our targeted augmentation strategy not only yields stronger accuracy gains but also does so with substantially lower computational cost.

### F.4 OBJECT DETECTION EXPERIMENTS

To further demonstrate the applicability of our method beyond image classification, we conducted experiments on object detection. We follow the setup of InstanceAugmentation (Kupyn & Rupprecht, 2024), which uses a pretrained ControlNet (Zhang et al., 2023)-based inpainter with prompt engineering to repaint individual objects. Following their data augmentation settings (Table 1 in (Kupyn & Rupprecht, 2024)), we trained a YOLOv5m model `https://github.com/ultralytics/yolov5` on the MS-COCO dataset (Lin et al., 2014) from scratch using default hyperparameters (batch size 40, 300 epochs). We used the synthetic images released by the authors of InstanceAugmentation. To identify slow-learnable images, we used the model checkpoint at epoch 5 and labeled any image containing at least one misclassified object as slow-learnable, resulting in 75% of the dataset.

We evaluate object detection performance using two standard metrics: $AP_{50}$ and $mAP_{50-95}$. Both metrics rely on the Intersection over Union (IoU), which measures how much a predicted bounding box overlaps with the ground truth. $AP_{50}$ computes Average Precision at an IoU threshold of 0.50, providing a lenient measure of detection accuracy—models with high $AP_{50}$ effectively locate objects even if their boxes are not perfectly aligned. In contrast, $mAP_{50-95}$ averages AP across IoU thresholds from 0.50 to 0.95 in steps of 0.05, rewarding models that produce both correct detections and tightly aligned bounding boxes. This makes $mAP_{50-95}$ a more stringent and comprehensive metric for evaluating detection quality.

Table 10 shows that our method also applies effectively to object detection, outperforming all baselines, including InstanceAugmentation while achieving a 25% reduction in data usage.

