# OpenReview forum: "Do We Need All the Synthetic Data? Targeted Image Augmentation via Diffusion Models"
_ICLR.cc/2026/Conference — ICLR 2026 Poster_

### Official Review · Reviewer_prBk · 2025-10-16

**Soundness:** 3
**Presentation:** 3
**Contribution:** 3
**Rating:** 6
**Confidence:** 5

**Summary:**

This author studies diffusion-based synthetic data augmentation for image classification. Unlike prior works that augment the entire dataset (often by generating 10×–30× more images), the authors propose targeted augmentation: identify the subset of training samples whose features are “slow-learnable” and generate synthetic variants only for these.

**Strengths:**

1) Focusing diffusion-based augmentation on slow-learnable examples is a fresh take on synthetic augmentation.
2)  The authors analyze a simplified two-layer CNN to compare SAM vs. SGD (showing SAM learns “noise” more slowly) and prove that generating faithful images accelerates learning of slow features without amplifying noise (Theorems 4.1–4.3).
3) The experiments are extensive and well-controlled. On CIFAR-10/100 and TinyImageNet, across multiple architectures, the proposed method consistently outperforms baselines (random subset, full augmentation, simple upsampling) and yields up to 2.8% test accuracy
4) By augmenting only ~30–40% of data, the method substantially reduces synthetic data generation time (e.g. 3.6h vs 12h on CIFAR-10) compared to full-data diffusion.

**Weaknesses:**

1) All experiments are on relatively small benchmarks (CIFAR-10/100, TinyImageNet). It is unclear if the approach scales to large datasets (e.g. full ImageNet) or real-world settings.
2) Identifying slow-learnable examples via early clustering of model outputs (or high loss) is somewhat heuristic.
3) The method relies on a text-conditional diffusion model (GLIDE) with class prompts and uses the real image as guidance. It is not fully clear how much the performance depends on prompt engineering or the specific diffusion backbone.
4) The 2-layer CNN analysis, while insightful, assumes a very stylized data distribution (two patches, Gaussian noise, cubic activations) and early-training approximations. It is not guaranteed these results carry over to deep architectures and natural images.

Missing relevant references:

1) GenMix: Effective Data Augmentation with Generative Diffusion Model Image Editing

2) Context-guided Responsible Data Augmentation with Diffusion Models

**Questions:**

1) How sensitive are the results to the specifics of the clustering step? For instance, does choosing a different number of clusters or a different layer for features change which examples are selected as “slow”?
2) Did you compare clustering versus simply picking top-θ% highest-loss examples (or uncertain examples)? Table 7 suggests clustering works better than high-loss, but can you elaborate on why?
3) How was the denoising step (e.g. 50 steps) chosen? Would an adaptive schedule (tailored per image) improve results?
4) Have you considered whether targeted augmentation helps other tasks (e.g. detection) or robustness measures beyond accuracy?
5) How does performance compare to simply oversampling the slow examples (as in weighted sampling) without generating new images? This would isolate the benefit of synthetic variety.

---

> ### Author Response · Authors · 2025-11-23
> **Response to Reviewer prBk (part 1)**
>
> We thank the reviewer for recognizing both theoretical and empirical strengths of our work. Below we address each concern in turn.
>
> ---
>
> **Scalability to Larger Datasets**
>
> - **ImageNet dataset**. In response to this concern, we have trained ResNet18 and ResNet50 on ImageNet from scratch using the ffcv library [1]. We applied our method to augment slow-learnable with synthetic images from Boomerang [2] which is a recent diffusion pipeline suggested by **Reviewer 9nHM**. The table below demonstrates that our method scales to ImageNet, outperforming all baselines including Boomerang while being 35% more efficient.
>
> | Method | Aug (%) | ResNet18 |  | ResNet50 |  |
> | --- | --- | --- | --- | --- | --- |
> |  |  | Top-1 | Top-5 | Top-1 | Top-5 |
> | Original | 0 | 67.23 $\pm$ 0.02 | 87.76 $\pm$ 0.10 | 73.24 $\pm$ 0.02 | 91.63 $\pm$ 0.08 |
> | Upsample | 65 | 68.86 $\pm$ 0.20 | 88.68 $\pm$ 0.12 | 74.04 $\pm$ 0.15 | 92.06 $\pm$ 0.08 |
> | Boomerang | 100 | 68.95 $\pm$ 0.04 | 89.02 $\pm$ 0.09 | 73.72 $\pm$ 0.06 | 92.09 $\pm$ 0.11 |
> | Ours + Boomerang | 65 | **69.28 $\pm$ 0.06** | **89.21 $\pm$ 0.09** | **74.77 $\pm$ 0.05** | **92.62 $\pm$ 0.07** |
>
> [1] Leclerc, Guillaume, et al. "FFCV: Accelerating training by removing data bottlenecks." Proceedings of the IEEE/CVF Conference on Computer Vision and Pattern Recognition. 2023.
>
> [2] Luzi, Lorenzo, et al. "Boomerang: Local sampling on image manifolds using diffusion models." arXiv preprint arXiv:2210.12100 (2022).
>
> - **More robust metric**. In our ImageNet experiments above, we also report top-5 accuracy, which is more robust than top-1 accuracy. Top-5 accuracy provides a more comprehensive evaluation in settings with many classes or high semantic overlap such as ImageNet because it captures cases where the model’s predictions are close but not exact.
>
> ---
>
> **Identify slow-learnable examples**
>
> - **Clustering is theoretically grounded and works better than other heuristics.** Our proposed clustering method is motivated by our theoretical analysis. Our method clusters the model outputs (logits), which according to our analysis capture the “total” alignment of model weights with different features. Early in training, only fast-learnable features are learned and thus have a large alignment with model weights. Therefore, only examples with fast-learnable features in a class have similar logits early in training. Thus, clustering the logits into two clusters allows separating slow- and fast-learnable examples effectively. More detailed discussion can be found in Sec 4.2 (lines 229-238).
>
>     Empirically, this approach performs better than alternatives such as using misclassification or loss (top-θ% highest-loss) with the same number of augmented images (see Table 7). As the loss for slow-learnable features oscillates (i.e., repeatedly gets low and high values) during the training (in particular early in training), separating slow- and fast-learnable examples by clustering the model outputs (which is more stable) is more robust than using loss or relying on misclassification.
>
> - **Choice of number of clusters**. We ran new experiments with different numbers of clusters and, for each configuration, varied how many clusters to augment—specifically selecting the clusters with the highest average loss. As shown in the table below and in Figure 9 of our revision, for CIFAR-10 with ResNet-18, our method which uses only two clusters and augments roughly 30% of examples yields the best performance. Since our objective is simply to separate examples into *slow* vs. *fast* groups, using two clusters is both conceptually natural and empirically the most effective.
>
>     For the choice of layer, according to our analysis the output layer captures the total alignment of features with mode weights more precisely. Thus, using the model output for clustering is a theoretically grounded choice.
>
>
> | #clusters | #augmented clusters | #augmented examples (%) | Test error (%) |
> | --- | --- | --- | --- |
> | 2 | 0 | 0% | 5.07 $\pm$ 0.04 |
> | 2 | 1 | 30% | **4.57 $\pm$ 0.02** |
> | 2 | 2 | 100% | 4.78 $\pm$ 0.04 |
> | 3 | 1 | 11% | 4.81 $\pm$ 0.05 |
> | 3 | 2 | 33% | 4.66 $\pm$ 0.06 |
> | 3 | 3 | 100% | 4.78 $\pm$ 0.04 |
> | 4 | 1 | 9% | 4.88 $\pm$ 0.08 |
> | 4 | 2 | 24% | 4.68 $\pm$ 0.06 |
> | 4 | 3 | 38% | 4.58 $\pm$ 0.05 |
> | 4 | 4 | 100% | 4.78 $\pm$ 0.04 |

---

> > ### Author Response · Authors · 2025-11-23
> > **Response to Reviewer prBk (part 2)**
> >
> > **Synthetic data generation pipeline**
> >
> > - **Prompt engineering**. Our method **does not** rely on prompt engineering. We intentionally use the simplest possible prompt (“a photo of <class name>”) without any manual tuning or optimization. This design choice highlights that our contributions lie in identifying which examples to augment, rather than in crafting prompts for image diversity and faithfulness. While richer or more descriptive prompts (e.g., conditional prompts used in DiffuseMix in Table 2) may further improve image quality, this is orthogonal to our framework. Our results demonstrate that even with minimal prompts, selectively augmenting the slow-learnable subset yields substantial improvements.
> > - **Generation backbone and method**. Our core contribution is independent of the specific diffusion model or editing pipeline. The success of our method comes from selecting which samples to augment—not from a particular choice of diffusion backbone. Our method can leverage any diffusion backbone to generate synthetic images by adding noise and denoising the original images. Empirically, we show in Tables 2 and 8 that our method works with diverse image-generation backbones, including DiffuseMix (InstructPix2Pix) and Boomerang (Stable Diffusion v1.5). More complex pipelines, such as GenMix and DiffCoRe-Mix, could also replace our simple GLIDE-based image variation and potentially yield further gains. This reinforces that our approach is generator-agnostic and complements advances in synthetic data generation.
> >
> >     We have added the discussion of GenMix and DiffCoRe-Mix to our revision. Thank you for bringing these to our attention.
> >
> > - **Number of denoising steps**. As shown in Table 6, we evaluated several denoising schedules and found that 50 steps provide the best trade-off between model performance and computational cost. Although adaptive, per-image noise schedules may offer additional gains, they also introduce nontrivial computational overhead. Exploring such adaptive strategies is an interesting direction for future work.
> > - **Oversampling slow examples**. This is a good question and can be confirmed in two ways. First, we use the UPSAMPLE baseline in all our experiments, which simply oversamples slow-learnable examples without generating any synthetic images. As shown in Table 5, UPSAMPLE consistently underperforms our synthetic augmentation across all upsampling factors k.
> >
> >     Second, for amplification factors larger than 2, e.g. for k = 5, in our original submission we reused synthetic images generated at different denoising steps (40/50/70/80) as shown in Table 4 to avoid rerunning diffusion multiple times. We additionally ran a new experiment where we used the same synthetic images (step 50) and repeated them four times. This variant yields a test error of 4.68, noticeably worse than our design. These results reinforce our theory (Section 4.2): diverse noise improves generalization, whereas repeated noise harms it, which explains why simple oversampling is less effective.
> >
> >
> >     The table below summarizes the test error of ResNet18 on CIFAR10 of three different strategies.
> >
> > | UPSAMPLE | Ours (same steps) | Ours (different steps) |
> > | --- | --- | --- |
> > | 4.98 | 4.68 | 4.42 |

---

> > > ### Author Response · Authors · 2025-11-23
> > > **Response to Reviewer prBk (part 3)**
> > >
> > > **Theoretical analysis vs practical architectures:**
> > >
> > > Deeper and more complex architectures that are truly reflective of practical deep learning are generally not theoretically tractable due to extreme nonlinearity, and richness of real images cannot be captured by statistical models in theory. Hence, consistent with prior work (Allen-Zhu & Li, 2020; Chen et al., 2022; Jelassi & Li, 2022; Cao et al., 2022; Kou et al., 2023; Deng et al., 2023; Chen et al., 2023), we analyzed a stylized feature-noise patch data model, which resembles real images, and a 2-layer CNN with cubic activation, which has sufficient nonlinearity but is still tractable. Similar or even simpler settings are used to show other properties of SAM, including smaller Hessian spectra (Foret et al., 2020; Kaur et al., 2023; Wen et al., 2022; Bartlett et al., 2023), sparser solution (Andriushchenko & Flammarion, 2022), benign overfitting in presence of weaker signal (Chen et al., 2022), and more uniform feature learning (Nguyen et al., 2024). See lines 101-110 for more detailed discussion. Despite analyzing a simpler model, we believe our insights carry over to more complex settings.
> > >
> > > Notably, based on the above simplified analysis **we proposed a method that’s generally applicable to any deep neural network**. To identify the examples containing the slow-learnable features, we train the model for a few epochs and find the cluster of examples with a high average loss, which contains examples that are not learned quickly by the model. This is generally applicable to any deep model to find slow-learnable features. Generating faithful synthetic data corresponding to these examples allows the model to learn the slow-learnable-features faster and improves the generalization. Our analysis on various deep models and datasets confirm the generality of our findings.

---

> > > > ### Comment · Reviewer_prBk · 2025-11-23
> > > > **Response to Author**
> > > >
> > > > Thank you for answering my questions. I have carefully reviewed the responses and the new experiments. Since all my concerns have been thoroughly addressed, I am increasing my rating from 6 to 8.

---

> > > > > ### Author Response · Authors · 2025-11-23
> > > > > **Thank you**
> > > > >
> > > > > Thank you very much for taking the time to review our work and for re-evaluating it. We sincerely appreciate your thoughtful questions and your careful assessment of our responses and new experiments. We’re glad that we were able to address all your concerns, and we truly appreciate your decision to increase your rating from 6 to 8.

---

### Official Review · Reviewer_9nHM · 2025-10-27

**Soundness:** 3
**Presentation:** 2
**Contribution:** 2
**Rating:** 4
**Confidence:** 3

**Summary:**

This paper proposes a targeted augmentation approach using diffusion models, focusing on "slow-learnable" images (identified via early training clustering) by adding noise and denoising to create faithful variations with different noise. A theoretical analysis with a two-layer CNN suggests this promotes uniform feature learning and reduces minibatch variance compared to upsampling. Experiments on CIFAR-10/100 and TinyImageNet show accuracy improvements (up to 2.8%) with ResNet, ViT, and other models, and the method complements optimizers like SGD and SAM.

**Strengths:**

**Efficiency**: Augmenting only 30–40% of data outperforms full-dataset augmentation, offering a practical, resource-aware solution.

**Empirical Support**: Ablation studies on augmentation factors and initialization provide useful insights.

**Compatibility**: Works well with existing methods (e.g., SAM), boosting performance further.

**Weaknesses:**

**Missing Prior**: The method overlaps with "Boomerang" [1], which uses similar noise-add-and-denoise techniques for data augmentation for classification, but it’s not cited or compared. Notably, they use all of the dataset for synthetic data generation, and they see gains in accuracy, which contradict experiments in this paper.

**Theory-Practice Gap**: The claim of mimicking SAM’s feature learning (e.g., sections 4.1–4.2 suggest SAM-like noise suppression and uniform learning) doesn’t fully align with empirical results, where gains add to SAM’s effects (abstract notes up to 2.8% improvement with SAM). This suggests the method might address different aspects of training dynamics than intended, and further analysis could clarify this discrepancy.

**Convergence claim**: The assertion of faster SGD convergence (Theorem 4.3, Corollary 4.4) relies on synthetic noise variance being lower than upsampling variance, but the link to the "small noise" assumption (section 4.4) isn’t fully derived or supported with training curves, leaving uncertainty about its practical impact.

**Idealized and unrealistic theory setting**: The model assumes simplified conditions (e.g., P=2 patches, orthogonal features in section 3), which may not capture the complexity of real image data, potentially limiting the theory’s applicability to broader settings.

**Scope limitation**: Experiments are confined to small datasets, and the claim of effectiveness across diffusion models (abstract) lacks support from multiple generators, which could restrict the method’s generalizability and leave its robustness untested.

[1] Luzi L, Mayer PM, Casco-Rodriguez J, Siahkoohi A, Baraniuk R. Boomerang: Local sampling on image manifolds using diffusion models. Transactions on Machine Learning Research.

**Questions:**

- Could you explain why the atypical activation function $\sigma(z) = z^3$ was chosen over ReLU, and does the theory hold if ReLU is used instead?

- Please cite, and compare/discuss results in the 'boomerang' paper, as it seems to contradict results in this paper.

-  Can authors please add training-loss curves to support the convergence claim and explore why stacking with SAM works?

- It would be great (but no necessary) to include a simple metric (e.g., feature similarity) to verify "faithfulness" of synthetic images.

- Could authors test on a larger dataset and with another diffusion model to broaden applicability?

## Overall

This is a helpful and promising approach for efficient augmentation, with solid small-scale results. Addressing the prior comparison, clarifying theory-practice links, and expanding experiments could make it even more impactful!

---

> ### Author Response · Authors · 2025-11-23
> **Response to Reviewer 9nHM (part 1)**
>
> We thank the reviewer for identifying the efficiency, compatibility and effectiveness of our method. We address the main concerns in turn:
>
> ---
>
> **Boomerang:**
>
> Please note that generating synthetic data by adding noise and denoising is not the main contribution of our work. As we discussed in Sec. 4.4, this approach was originally introduced by (He et al., 2023), which we properly cited in line 339. Boomerang proposal is a more recent variant of (He et al. 2023) and thus can be easily applied with our method.
>
> Importantly, we would like to clarify the following points which may have been misunderstood by the reviewer:
>
> - **We do not claim that full-data augmentation degrades the performance. We show that targeted data augmentation (augmenting only a part of the data) outperforms full-data augmentation**. To our knowledge, our work is the first to show this interesting result.
> - Therefore, our findings are not contradictory to Boomerang. Indeed, **using our approach to augment a smaller part of data with Boomerang outperforms full-data augmentation by Boomerang**, as we confirm with our new experiments below.
>
> | Method | Aug (%) | ResNet18 |  | ResNet50 |  |
> | --- | --- | --- | --- | --- | --- |
> |  |  | Top-1 | Top-5 | Top-1 | Top-5 |
> | Original | 0 | 67.23 $\pm$ 0.02 | 87.76 $\pm$ 0.10 | 73.24 $\pm$ 0.02 | 91.63 $\pm$ 0.08 |
> | Upsample | 65 | 68.86 $\pm$ 0.20 | 88.68 $\pm$ 0.12 | 74.04 $\pm$ 0.15 | 92.06 $\pm$ 0.08 |
> | Boomerang | 100 | 68.95 $\pm$ 0.04 | 89.02 $\pm$ 0.09 | 73.72 $\pm$ 0.06 | 92.09 $\pm$ 0.11 |
> | Ours + Boomerang | 65 | **69.28 $\pm$ 0.06** | **89.21 $\pm$ 0.09** | **74.77 $\pm$ 0.05** | **92.62 $\pm$ 0.07** |
>
> In our revision we have now added discussions of Boomerang to our Related works and comparison with Boomerang on ImageNet to Appendix F3.
>
> ---
>
> **Theory-practice gap:**
>
> It appears there might be a misunderstanding of our results. In sec 4.1, 4.2, we show that SAM suppresses learning noise and learns features more uniformly, compared to GD. This does not mean that SAM doesn’t learn any noise and learns all the features at the same time. This means that compared to GD, SAM learns less noise and learns features at a more uniform speed. Our synthetic data augmentation ensures that gradient-based optimizers (e.g. SAM or GD) learn less noise and learn features more uniformly. Therefore, our method applied to SAM further reduces the amount of noise learned by SAM and allows learning features at a more uniform speed. Thus, our method improves both GD and SAM. This is the cumulative effect not a contradictory finding.
>
> ---
>
> **Convergence claim**:
>
> In Figure 7 (original submission), we do empirically observe that the gradient variance of synthetic data is consistently lower than upsampled data. Figure 8 (new in revision) in the Appendix shows that training on synthetically augmented data achieves a lower loss than training on the upsampled data. This confirms the validity of our assumption in Sec 4.4 (c.f. Lines 1516-1524 for the discussion).
>
> ---
>
> **Idealized and unrealistic theory setting**:
>
> Deeper and more complex architectures that are truly reflective of practical deep learning are generally not theoretically tractable due to extreme nonlinearity, and richness of real images cannot be captured by statistical models in theory. Hence, consistent with prior work (Allen-Zhu & Li, 2020; Chen et al., 2022; Jelassi & Li, 2022; Cao et al., 2022; Kou et al., 2023; Deng et al., 2023; Chen et al., 2023), we analyzed a stylized feature-noise patch data model, which resembles real images, and a 2-layer CNN with cubic activation, which has sufficient nonlinearity but is still tractable. Similar or even simpler settings are used to show other properties of SAM, including smaller Hessian spectra (Foret et al., 2020; Kaur et al., 2023; Wen et al., 2022; Bartlett et al., 2023), sparser solution (Andriushchenko & Flammarion, 2022), benign overfitting in presence of weaker signal (Chen et al., 2022), and more uniform feature learning (Nguyen et al., 2024). See lines 101-110 for more detailed discussion. Despite analyzing a simpler model, we believe our insights carry over to more complex settings.
>
> Notably, based on the above simplified analysis **we proposed a method that’s generally applicable to any deep neural network**. To identify the examples containing the slow-learnable features, we train the model for a few epochs and find the cluster of examples with a high average loss, which contains examples that are not learned quickly by the model. This is generally applicable to any deep model to find slow-learnable features. Generating faithful synthetic data corresponding to these examples allows the model to learn the slow-learnable-features faster and improves the generalization. Our analysis on various deep models and datasets confirm the generality of our findings.

---

> > ### Author Response · Authors · 2025-11-23
> > **Response to Reviewer 9nHM (part 2)**
> >
> > **Activation function**:
> >
> > Using cubic activation is a common choice in theoretical analysis because smooth, continuously differentiable functions like cubic make the analysis easier. Similar to (Allen-Zhu & Li, 2020; Chen et al., 2022; Jelassi & Li, 2022; Cao et al., 2022; Kou et al., 2023; Deng et al., 2023; Chen et al., 2023), we adopt this choice for theoretical simplicity. However, we believe that the theory still holds if we use ReLU, but requires more sophisticated techniques like those in (Chen et al., 2023).
> >
> > ---
> >
> > **Scope of experiments:**
> >
> > - **Multiple generators**: As shown in Tables 2 and 8, our method also works effectively with synthetic images produced by different diffusion pipelines including DiffuseMix (InstructPix2Pix) and Boomerang (Stable Diffusion v1.5) in addition to GLIDE.
> > - **Large-scale datasets**: We trained ResNet18 and ResNet50 on ImageNet from scratch using the ffcv library [1]. Our method outperforms all baselines including Boomerang while being 35% more efficient.
> > - **Faithfulness of synthetic images**: We have already calculated the feature similarity which is FID score in Table 6 in our Appendix. While synthetic examples should have similar features to real data, having too similar features resembles upsampling and does not yield optimal performance, as we confirmed in all our experiments.
> >
> > [1] Leclerc, Guillaume, et al. "FFCV: Accelerating training by removing data bottlenecks." Proceedings of the IEEE/CVF Conference on Computer Vision and Pattern Recognition. 2023.

---

> > > ### Comment · Reviewer_9nHM · 2025-11-24
> > > **Response to authors**
> > >
> > > Thank you for your response, the new manuscript and authors response answers majority of my concerns. I will raise my score.

---

> ### Author Response · Authors · 2025-11-24
> **Thank you**
>
> We appreciate you taking the time to review our revised manuscript and responses. We’re glad to hear that the updates addressed the majority of your concerns, and we sincerely appreciate your decision to raise your score from 4 to 6. Thank you also for suggesting Boomerang—adding it significantly strengthened our paper.

---

### Official Review · Reviewer_s4Wo · 2025-10-31

**Soundness:** 3
**Presentation:** 2
**Contribution:** 2
**Rating:** 4
**Confidence:** 3

**Summary:**

This paper proposes a novel data augmentation strategy designed to improve the generalization of image classification models. The core contribution is a method that selectively applies augmentation only to a subset of the training data identified as slow-learning samples. The authors demonstrate that this targeted approach improves classification performance.

**Strengths:**

- The central idea of targeting slow-learning samples for augmentation is novel and intuitive. The rationale that focusing augmentation efforts on more challenging examples seems a logical approach to improving model robustness and generalization.
- The paper provides extensive empirical validation across three different datasets, showing credibility to the proposed method's effectiveness. The observation regarding the characteristics of slow-learned samples is particularly interesting and further discussion on this would make paper more interesting.

**Weaknesses:**

- The theoretical analysis relies on a simplified two-layer CNN assumption. This raises questions about the direct applicability and relevance of the derived theorems to the deeper, more complex architectures commonly used in practice. The paper would be strengthened by a discussion bridging this theoretical gap.
- I have concerns regarding the significant computational overhead of the proposed method. Utilizing a diffusion model for data generation, even for a subset of the data, is inherently more expensive than traditional augmentation techniques. Also, the multi-step pipeline may limit its practical adoption.
- The experiments are confined to relatively small-scale datasets. It is unclear how the method would perform on larger, more complex datasets such as ImageNet. An explanation for the choice of datasets and a discussion on the method's potential scalability would be beneficial.

**Questions:**

- What are the fundamental, identifiable differences between the samples classified as "slow-learning" versus "fast-learning"? If distinct features or patterns characterize these slow-learning samples, could a model be developed to identify them a priori? Such an approach could simplify the overall pipeline by removing the need for an initial training phase solely to identify these samples.
- Could the proposed augmentation strategy, which focuses on difficult examples, be adapted to benefit tasks outside of classification, such as improving sample quality or diversity in image generation?

---

> ### Author Response · Authors · 2025-11-23
> **Response to Reviewer s4Wo (part 1)**
>
> We thank the reviewer for the thoughtful feedback and for recognizing the novelty of our approach and the breadth of empirical experiments. Below we will address the main concerns:
>
> ---
>
> **Theoretical analysis vs practical architectures:**
>
> Deeper and more complex architectures that are truly reflective of practical deep learning are generally not theoretically tractable due to extreme nonlinearity, and richness of real images cannot be captured by statistical models in theory. Hence, consistent with prior work (Allen-Zhu & Li, 2020; Chen et al., 2022; Jelassi & Li, 2022; Cao et al., 2022; Kou et al., 2023; Deng et al., 2023; Chen et al., 2023), we analyzed a stylized feature-noise patch data model, which resembles real images, and a 2-layer CNN with cubic activation, which has sufficient nonlinearity but is still tractable. Similar or even simpler settings are used to show other properties of SAM, including smaller Hessian spectra (Foret et al., 2020; Kaur et al., 2023; Wen et al., 2022; Bartlett et al., 2023), sparser solution (Andriushchenko & Flammarion, 2022), benign overfitting in presence of weaker signal (Chen et al., 2022), and more uniform feature learning (Nguyen et al., 2024). See lines 101-110 for more detailed discussion. Despite analyzing a simpler model, we believe our insights carry over to more complex settings.
>
> Notably, based on the above simplified analysis **we proposed a method that’s generally applicable to any deep neural network**. To identify the examples containing the slow-learnable features, we train the model for a few epochs and find the cluster of examples with a high average loss, which contains examples that are not learned quickly by the model. This is generally applicable to any deep model to find slow-learnable features. Generating faithful synthetic data corresponding to these examples allows the model to learn the slow-learnable-features faster and improves the generalization. Our analysis on various deep models and datasets confirm the generality of our findings.
>
> ---
>
> **Computational overhead and practicality:**
>
> 1.
> **Motivation.** We agree that generating synthetic data with diffusion models is very expensive and reducing the high costs of synthetic data augmentation is exactly the motivation of our method. These methods are popular as they often outperform traditional data augmentation methods and achieve SOTA performance. However, they increase the size of training data by up to 10-30x to achieve desirable performance.
>
> **Contribution.** Our work does not introduce a new diffusion generation method. The main contribution of our work is to show, for the first time, that synthetically augmenting 30%-40% of the examples significantly outperforms full data augmentation and is considerably more efficient. We showed: (1) what are these examples, i.e. those that are learned later in training, and (2) how they should be augmented, i.e. by preserving the feature and distorting noise, which can be done considerably faster than generating images from pure noise (as is done many standard diffusion pipelines).
>
> **Application to non-diffusion augmentation methods**. Our results on “which examples should be augmented” can be applied to other (e.g. traditional) data augmentation methods. To confirm this, we conducted new experiments where we applied TrivialAugment (Müller & Hutter, 2021)--which is a very popular non-diffusion data augmentation method--to **only** augment real slow-learnable examples and compared it with using TA applied to full data. We see that augmenting the slow-learnable examples with TA outperforms augmenting the full data with TA. In addition, augmenting the slow-learnable examples with TA and synthetic diffusion augmentation further boost the performance and outperforms augmenting full data with TA and synthetic data augmentation. The table below demonstrates the test error when training ResNet18 on CIFAR10.
>
> | Dataset | Test error |
> | --- | --- |
> | Original | 5.07 $\pm$ 0.04 |
> | Augmenting only slow-learnable examples with TA | 4.26 $\pm$ 0.02 |
> | Augmenting all examples with TA | 4.48 $\pm$ 0.08 |
> | Augmenting only slow-learnable examples with TA and synthetic data with diffusion model | 3.81 $\pm$ 0.10 |
> | Augmenting all examples with TA and synthetic data with diffusion model | 4.20 $\pm$ 0.13 |

---

> > ### Author Response · Authors · 2025-11-23
> > **Response to Reviewer s4Wo (part 2)**
> >
> > **Computational overhead and practicality:**
> >
> > 2. Our pipeline is actually very simple and light weight and its additional cost is negligible compared to the cost of synthetic data augmentation. Specifically, we train the model for a few epochs (e.g. 4 epochs in a 200 epoch training pipeline) and partition the model outputs into two clusters (the cost of clustering is negligible). We generate synthetic data for only the cluster with larger average loss (containing only 30%-40% of the examples), instead of the full data. Synthetic data generation is much more expensive compared to training the model for a few epochs. Thus, our entire pipeline is much faster compared to synthetically augmenting the full data, which is done by the baselines.
> > 3. Please refer to our Appendix for reported training times and a comprehensive comparison between different augmentation methods (Table 3).
> >
> > ---
> >
> > **Depth of Experiments**
> >
> > Our experiments follow the choices of models and datasets from the original SAM paper and covers several more complex architectures (including VGG19, DenseNet121, ViT-S, ConvNeXt, Swin-T) and more datasets (Tiny-ImageNet, Flowers-102, Aircraft, Stanford Cars).
> >
> > To answer your question on how our method scales to larger datasets, we have trained ResNet18 and ResNet50 on ImageNet from scratch using the ffcv library [1]. We applied our method to augment slow-learnable with synthetic images from Boomerang [2] which is a recent diffusion method suggested by **Reviewer 9nHM**. The table below demonstrates that our method scales to ImageNet, outperforming all baselines including Boomerang while being 35% more efficient.
> >
> > | Method | Aug (%) | ResNet18 |  | ResNet50 |  |
> > | --- | --- | --- | --- | --- | --- |
> > |  |  | Top-1 | Top-5 | Top-1 | Top-5 |
> > | Original | 0 | 67.23 $\pm$ 0.02 | 87.76 $\pm$ 0.10 | 73.24 $\pm$ 0.02 | 91.63 $\pm$ 0.08 |
> > | Upsample | 65 | 68.86 $\pm$ 0.20 | 88.68 $\pm$ 0.12 | 74.04 $\pm$ 0.15 | 92.06 $\pm$ 0.08 |
> > | Boomerang | 100 | 68.95 $\pm$ 0.04 | 89.02 $\pm$ 0.09 | 73.72 $\pm$ 0.06 | 92.09 $\pm$ 0.11 |
> > | Ours + Boomerang | 65 | **69.28 $\pm$ 0.06** | **89.21 $\pm$ 0.09** | **74.77 $\pm$ 0.05** | **92.62 $\pm$ 0.07** |
> >
> > [1] Leclerc, Guillaume, et al. "FFCV: Accelerating training by removing data bottlenecks." Proceedings of the IEEE/CVF Conference on Computer Vision and Pattern Recognition. 2023.
> >
> > [2] Luzi, Lorenzo, et al. "Boomerang: Local sampling on image manifolds using diffusion models." arXiv preprint arXiv:2210.12100 (2022).
> >
> > ---
> >
> > **Regarding the questions:**
> >
> > **Q1.** Figure 1 shows examples of slow-learnable and fast-learnable examples. Fast-learnable images are clearly representative of their classes. On the other hand slow-learnable images are visually less identifiable, with blurry edges and ambiguous targets. Nevertheless, exact patterns are dataset-specific.
> >
> > Our method proposes an unsupervised approach to separate slow-learnable features. While in general, it might be possible to train a classifier to separate fast and slow-learnable features, this approach has two challenges: (1) it requires labels for fast and slow-learnable examples (which can be obtained e.g. with our method) and (2) since different models (e.g. ResNet50 vs CNN) learn features with different speed, for accurate classification such labels should be obtained for each model separately. Our approach is simple and lightweight (the cost of training a model for a few epochs is often negligible), and thus we didn’t explore this direction. But, this can be an interesting direction for future work.
> >
> > Finally, we note that the slow-learnable subset obtained by our method transfers across similar architectures. For example, when training ResNet18 on the synthetic-augmented CIFAR-10 dataset selected by VGG19 with k = 2, the test error is $4.57 \pm 0.10$, matching the performance achieved when using samples selected by ResNet18 itself $4.57 \pm 0.02$. Thus, in practice we can find the slow-learnable examples once and use them for other similar models.
> >
> > **Q2.** This is an interesting question. While our theory only analyzes classification tasks, we expect our method to benefit diversity and quality of other tasks, such as image generation. Our method generates synthetic examples for the part of the data that is not learned early in training to amplify the features that are less represented in the data. In doing so, it effectively improves the diversity of the training dataset. For image generation, a more diverse training dataset results in generating more diverse images. We plan to set up and apply our method to other vision tasks and will report the results if we get them in time. Thank you for the suggestion!

---

### Official Review · Reviewer_pN7D · 2025-10-31

**Soundness:** 3
**Presentation:** 3
**Contribution:** 3
**Rating:** 6
**Confidence:** 5

**Summary:**

This paper proposed a training strategy that can smartly combine real data and synthetic data for training an improved classifier. To verify the effectiveness, this paper evaluated the method on image data augmentation for classification across backbones and datasets. This paper also provided some analysis on simple MLP layers. Besides, the method is a plug-and-play module and also evaluated plugged into the DiffuseMix.

**Strengths:**

The dominant strength is that the current data augmentation paper only focuses on how to generate data with high fidelity and diversity for a more robust decision boundary. However, a very small paper focuses on how to balance the real set and the synthetic set during the training process. This paper fills the blank for current generative-based data augmentation research.

**Weaknesses:**

This method is general, but the evaluations are limited.

1/ The evaluated backbones are too weak, and whether better-pretrained backbones can overlay the benefit of your method.

2/ Since this method is a plug-and-play module, why not evaluate it based on more state-of-the-art methods like [1,2,3,4]? Meanwhile, you should at least discuss them in the related work.

3/ Lack of evaluations on fine-grained datasets.

4/ This method seems like can be applied not only for image classification datasets, how for the augmentations in detection, segmentation even in other modalities like text and videos.


References

[1] Effective Data Augmentation With Diffusion Models

[2] Enhance image classification via inter-class image mixup with diffusion model

[3] Inversion Circle Interpolation: Diffusion-based Image Augmentation for Data-scarce Classification

[4] Advancing Fine-Grained Classification by Structure and Subject Preserving Augmentation

**Questions:**

If you can solve my concerns, the method can be very general, and then I can raise my score to 8.

---

> ### Author Response · Authors · 2025-11-23
> **Response to Reviewer pN7D**
>
> We thank the reviewer for their valuable feedback and recognizing that our work fills a gap in current generative-based data augmentation research. Below we address each concern in turn.
>
> ---
>
> **Stronger backbones**
>
> Our method can be applied to any pretrained backbones. In fact, from Table 2 in Section 5.1, we have conducted transfer learning experiments, in which we used ResNet18 pretrained on ImageNet as the backbone. In addition, we have conducted new experiments where we apply our method to training ResNet50 on ImageNet. We used a recent diffusion pipeline, namely Boomerang [5] suggested by **Reviewer 9nHM**. The table below illustrates that synthetically augmenting 65% of slow-learnable images with Boomerang outperforms full-data generation with Boomerang. Even in this stronger setting, our targeted augmentation method still outperforms baselines with substantial improvements.
>
> | Method | Aug (%) | ResNet18 |  | ResNet50 |  |
> | --- | --- | --- | --- | --- | --- |
> |  |  | Top-1 | Top-5 | Top-1 | Top-5 |
> | Original | 0 | 67.23 $\pm$ 0.02 | 87.76 $\pm$ 0.10 | 73.24 $\pm$ 0.02 | 91.63$\pm$ 0.08 |
> | Upsample | 65 | 68.86 $\pm$ 0.20 | 88.68 $\pm$0.12 | 74.04 $\pm$ 0.15 | 92.06 $\pm$ 0.08 |
> | Boomerang | 100 | 68.95 $\pm$ 0.04 | 89.02 $\pm$ 0.09 | 73.72 $\pm$ 0.06 | 92.09 $\pm$ 0.11 |
> | Ours + Boomerang | 65 | **69.28 $\pm$ 0.06** | **89.21 $\pm$ 0.09** | **74.77 $\pm$ 0.05** | **92.62 $\pm$ 0.07** |
>
> ---
>
> **Other state-of-the-art methods**
>
> Thank you for your suggestion. We have already discussed [1] (Trabucco et al., 2023) in our Related Works section, and in our revision we have now added discussions of [2, 3, 4] as well. Our paper addresses “which examples should be synthetically augmented” and “what are the characteristics of high-quality synthetic examples”. Thus, it can be directly combined with state-of-the-art diffusion-based augmentation pipelines to generate faithful images. Our original experiments with DiffuseMix (in Table 2) and our new experiments above with Boomerang [5] show consistent performance improvements, further confirming the effectiveness of our method applied to state-of-the-art generation methods.
>
> [5] Luzi, Lorenzo, et al. "Boomerang: Local sampling on image manifolds using diffusion models." arXiv preprint arXiv:2210.12100 (2022).
>
> ---
>
> **Fine-grained datasets**
>
> We agree that fine-grained datasets would further validate the method. Beyond general image datasets like CIFAR and Tiny-ImageNet, our original submission already includes experiments on (1) Flowers-102, (2) Aircraft, and (3) Stanford Cars (which are also used in [1, 2, 3, 4]), where our approach consistently improves over original data and non-targeted synthetic baseline. This supports precisely the use case the reviewer has in mind and connects naturally to [4], which also focuses on fine-grained settings. We will make this connection to fine-grained classification more explicit in the text.
>
> ---
>
> **Extension to other domains:**
>
> This is a very interesting point. While our theory only analyzes classification tasks, we expect that our method benefits other tasks, such as detection and segmentation and other modalities like video and text. We are setting up and applying our method to some of the above tasks and will report the results if we get them in time. Thank you for the suggestion!
>
> ---
>
> We hope the above clarifications alleviate your concern. To recap, our empirical evaluation already spans:
>
> - **Multiple datasets of different granularity:** CIFAR-10, CIFAR-100, Tiny-ImageNet, ImageNet and three **fine-grained** datasets (Flowers-102, Aircraft, Stanford Cars);
> - **Multiple architectures:** VGG19, DenseNet121, ViT-S, ConvNeXt-T, Swin-T, ResNet18, ResNet50;
> - **Multiple optimizers:** SGD and SAM;
> - **Composition with a SOTA diffusion-based augmentation method (DiffuseMix, Boomerang)** to demonstrate plug-and-play compatibility.
>
> Taken together with our theoretical analysis, we believe this provides strong evidence that selectively augmenting slow-learnable examples is both effective and general within the image classification setting. We will improve the presentation to better highlight this breadth.
>
> Please let us know if we have addressed your concerns. We are happy to provide more experiments and clarify further.

---

### Author Response · Authors · 2025-12-03
**New object detection experiments**

To further demonstrate the applicability of our method beyond image classification, we conducted experiments on object detection. We followed the setup of InstanceAugmentation [1], which uses a pretrained ControlNet[2]-based inpainter with prompt engineering to repaint individual objects. Following their data augmentation settings (Table 1 in [1]), we trained a YOLOv5m model (https://github.com/ultralytics/yolov5) on the MS-COCO dataset [3] from scratch using default hyperparameters (batch size 40, 300 epochs). We used the synthetic images released by the authors of InstanceAugmentation.

To identify slow-learnable images for targeted augmentation, we used the checkpoint at epoch 5 without further tuning, as training YOLOv5m on the original MS-COCO already requires 1 day on 4 GPUs. We then labeled an image as slow-learnable if at least one object was misclassified under the model’s default inference pipeline. We chose misclassification as the criterion because it is straightforward to obtain in YOLO inference code, though our ablations (Table 7) suggest that clustering-based identification would likely yield even stronger performance. This procedure marked approximately 75% of the dataset as slow-learnable.

We evaluated object detection performance using two standard metrics: $\text{AP}\_{50}$ and $\text{mAP}\_{50-95}$. Both metrics rely on the Intersection over Union (IoU), which measures how much a predicted bounding box overlaps with the ground truth. $\text{AP}\_{50}$ computes Average Precision at an IoU threshold of 0.50, providing a lenient measure of detection accuracy—models with high $\text{AP}\_{50}$ effectively locate objects even if their boxes are not perfectly aligned. In contrast, $\text{mAP}\_{50-95}$ averages AP across IoU thresholds from 0.50 to 0.95 in steps of 0.05, rewarding models that produce both correct detections and tightly aligned bounding boxes. This makes $\text{mAP}\_{50-95}$ a more stringent and comprehensive metric for evaluating detection quality.

The table below shows that our method also applies effectively to object detection, outperforming all baselines, including InstanceAugmentation while achieving a 25% reduction in data usage.

| Method | Aug (%) | $\text{AP}_{50}$ | $\text{mAP}_{50-95}$ |
| --- | --- | --- | --- |
| Original | 0 | 63.26 | 44.26 |
| Upsample | 75 | 63.87 | 44.92 |
| InstanceAugmentation | 100 | 64.17 | 45.75 |
| Ours + InstanceAugmentation | 75 | **64.94** | **46.28** |

[1] Kupyn, Orest, and Christian Rupprecht. "Dataset enhancement with instance-level augmentations." European Conference on Computer Vision. Cham: Springer Nature Switzerland, 2024.

[2] Zhang, Lvmin, Anyi Rao, and Maneesh Agrawala. "Adding conditional control to text-to-image diffusion models." Proceedings of the IEEE/CVF international conference on computer vision. 2023.

[3] Lin, Tsung-Yi, et al. "Microsoft coco: Common objects in context." European conference on computer vision. Cham: Springer International Publishing, 2014.

---

### Author Response · Authors · 2025-12-03
**Summary of paper contributions and changes made to the revision**

Dear Area Chairs,

We appreciate your extra effort in reviewing our paper and rebuttal. We would like to summarize our contributions and provide a brief summary of the key changes we made to our revision.

- **Contributions**: Our paper is, to our knowledge, the **first work to *theoretically* prove and *empirically* demonstrate that selectively augmenting 30-40% of data points (containing features that are not learned early in training) significantly outperforms augmenting the full dataset**. This addresses a timely challenge in large-scale machine learning.
    - In particular, for the expensive SOTA synthetic data augmentation methods based on diffusion models, this reduces the time to generate synthetic data and training on the augmented data by up to 168 hours (on ImageNet), while resulting in 1% higher accuracy (using ResNet50).
    - Theoretically, our method improves generalization by promoting homogeneity in feature learning speed without amplifying noise in training data.
    - Empirically, our method for identifying which examples to augment is very light weight and can be applied to any SOTA (and traditional) data augmentation methods to improve their performance.
    - The novelty of our approach, its fresh take and effectiveness, and our extensive experiments are acknowledged by all the reviewers.
- **Changes to our revision (with blue font):** During the discussion phase, we conducted new experiments confirming the effectiveness of our method across:
    - ***Various SOTA data augmentation methods**,* including (1) synthetic data augmentation methods (GLIDE, DiffuseMix, Boomerang, and InstanceAugmentation) with different diffusion models (GLIDE, Instruct Pix2Pix, Stable Diffusion, and ControlNet), and (2) traditional data augmentation methods, including TrivialAugment, etc. This confirms the versatility of our method.
    - ***Larger-scale datasets and stronger backbones***, including training ResNet50 and ResNet18 on ImageNet, demonstrating that targeted augmentation outperforms full augmentation at large scale.
    - ***Other tasks beyond classification,*** including new object detection experiments on MS-COCO using YOLOv5, showing performance improvements over full augmentation, confirming the generality of our method and its application beyond classification.
- The settings of our theoretical analysis follow prior related works and we empirically verified our results **across a wide range of model architectures** (ResNet, VGG, DenseNet, ViT, ConvNeXt, Swin).
- We expanded our Related Works section to include additional related works suggested by reviewers.

Based on our rebuttal, Reviewer **prBk** and **9nHM** increased their scores to 8 and 6, respectively. Reviewer **pN7D** mentioned that they can increase their score to 8 if we confirm the applicability of our method to SOTA diffusion augmentation methods and beyond classification tasks, which we addressed in our rebuttal. While we haven’t heard back from Reviewer **s4Wo**, we’ve addressed all their questions in our rebuttal.

Thank you again for your valuable time and effort.

Best regards,

Authors

---

### Meta-Review · Area_Chair_EFQF · 2026-01-11

**Summary:**

The proposed targeted or partial augmentation method only alters a subset of the training images to bolster the training set with synthetic data. The experiments confirm a modest but consistent boost of ~2% absolute across models (ResNet, ViT, ConvNeXt, Swin) on the smaller datasets of CIFAR-10/100 and Tiny ImageNet. Theoretical analysis of a simplified 2-layer convnet supports the method.

Four expert reviewers with work on generative modeling and synthetic data are split between acceptance (pN7D: 6, prBk: 6) and rejection (s4Wo: 4, 9nHM: 4). The key factors for the decision raised by the reviews are 0. the place for this work in its focus on targeting augmentation and balancing real and synthetic data, 1. missing related work and comparisons for the most recent and best diffusion augmentation methods, 2. the limited scale of datasets and models in the experiments, 3. the limited scope for classification only, and 4. the gap between theory and experiment. The positive point 0. is justified by the rebuttal and further results and the negative points 1-4 have been resolved or tempered by the discussion and further results.

The rebuttal replies to each review and provides a general response. Key points are addressed by rebuttal experiments: scaling to a larger dataset and model with ImageNet and ResNet-50, the use of more recent and state-of-the-art diffusion augmentation methods like Boomerang, the application of the targeting method to computationally efficient augmentation like TrivialAugment, and a proof-of-concept result for object detection in addition to the existing results on image classification. The submission has already been revised to show how the changes requested by review can and have been incorporated.

The meta-reviewer sides with acceptance in consideration of the initial reviews, the rebuttal and its results, and the in-scope discussion comments along with the meta-reviewers own best inference of potential updates to ratings. While borderline as submitted, the rebuttal and revision push this work to the side of acceptance, because the contributed method is informative w.r.t. the state-of-the-art for diffusion augmentation and compatible with multiple datasets, models, and tasks.

Miscellaneous feedback: The meta-reviewer suggests naming the method, rather than relying on the generic "Ours", for ease of reference. This has no bearing on the decision.

**Reviewer Concerns:**

- Missing related work and comparisons from the state-of-the-art for generation and synthetic data augmentation (pN7D, 9nHM, prBk): This is resolved by including the latest and suggested references in the related work and adding Boomerang to the experiments alongside the existing results for DiffuseMix.
- Small-scale datasets and weak backbones (pN7D, 9nHM, s4Wo, prBk): This is resolved in the rebuttal by scaling to ImageNet and ResNet-50.
- Restricted application to only classification (pN7D, s4Wo, prBk): The rebuttal provides a first experiment showing compatibility with object detection. While positive, it is only one experiment, so this is partially resolved.
- Gap between theory and practice and the relevance of the theoretical setting (9nHM, s4Wo, prBk): There is indeed a disconnect between the simplified model for the theoretical content and the real models for the empirical content. However, this gap is not unusual due to the difficulty of fully mathematically characterizing learned deep networks as high-dimensional and nonlinear models, and this is compensated by the quantity and quality of empirical results justifying the method. If this were a purely theoretical paper, the issue of validity would be a fair concern, but in this case it serves as a potential insight that may apply to the real case. The rebuttal provides more discussion of this, and underlining the simplicity of the theoretical model vs. the real experiments in the revision would completely resolve this.
- Computational cost due to the expense of diffusion modeling (s4Wo): While the proposed use of diffusion is expensive, this is the case for existing methods that already rely on diffusion, so the partial augmentation of only a targeted subset of the data as proposed in this work can actually be seen as an improvement in efficiency. This is therefore resolved. Furthermore a rebuttal experiment shows the targeted selection of data to augment applies to more efficient augmentation techniques like TrivialAugment too, and the improvement in error compounds when combined with diffusion augmentation also.
- Oversampling slow examples vs. augmenting (prBk): This control experiment is covered by the "upsample" baseline in the paper, which could be better named as oversample for clarity, and by further discussion in the rebuttal.

**Reviewer Scores:**

- pN7D: would raise from 6 to 7 given the rebuttal and the new results provided that are specifically targeted at the requests for more backbones, combination with more recent methods, and further datasets and tasks. The proposed method does combine with more state-of-the-art methods and the references provided by the review have been incorporated into the related work.
- prBk: would raise from 6 to 7 or 8 given the rebuttal and its new results showing scalability to ImageNet plus the discussion. This follows from the points resolved and their own comment.
- s4Wo: would likely raise from 4 to at least 6 because their specific point about computational complexity is positively resolved and the shared points about scale and application beyond classification are also addressed by the rebuttal. Given the positive reaction of the other reviewers with related concerns, it is a reasonable to expect this reviewer could increase their score too.
- 9nHM: would raise from 4 to 6 given the review, rebuttal, and discussion. The referenced related work, Boomerang, has been incorporated in the revision and in a rebuttal experiment the proposed method and Boomerang combine well. The increase follows from these resolved points and their own comment.

---

### Decision · Program_Chairs · 2026-01-26

Accept (Poster)